

# Global estimates of 100-year return values of daily precipitation from ensemble weather prediction data

Florian Ruff and Stephan Pfahl

Freie Universität Berlin, Institute of Meteorology, Carl-Heinrich-Becker-Weg 6-10, 12165 Berlin, Germany

**Correspondence:** Florian Ruff (florian.ruff@met.fu-berlin.de)

**Abstract.** High-impact river floods are often caused by very extreme precipitation events with return periods of several decades or centuries, and the design of flood protection measures thus relies on reliable estimates of the corresponding return values. However, calculating such return values from observations is associated with large statistical uncertainties due to the limited length of observational time series. Here, 100-year return values of daily precipitation are estimated on a global grid based on a large data set of model-generated precipitation events from ensemble weather prediction. In this way, the statistical uncertainties of the return values can be substantially reduced compared to observational estimates. In spite of a general agreement of spatial patterns, the model-generated data set leads to systematically higher return values than the observations in many regions, with statistically significant differences, for instance, over the Amazon, western Africa, the Arabian Peninsula and India. This may point to an underestimation of very extreme precipitation events in observations, which, if true, would have important consequences for practical water management.

## 1 Introduction

Extreme precipitation is often associated with river floods, which are one of the most dangerous hazards for society and have caused high socio-economic losses around the globe (Merz et al., 2021; Kron, 2015; Barredo, 2007; Douben, 2006). There are many historical examples of these events such as several flash floods due to a mesoscale convective system in South America in 2022 (Alcântara et al., 2023), an extreme flood event in Central Europe in 2021 (Mohr et al., 2023) and a widespread flooding in Thailand during the monsoon season in 2011 (Gale and Saunders, 2013). River floods typically develop along one of two distinct pathways, either very quickly in form of a flash flood or as a slower increase of the runoff and water levels over several hours. On the one hand, flash floods, which often occur in smaller rivers, are associated with very high precipitation rates in a spatially limited domain, leading to a quickly developing peak discharge (Sene, 2016) that can threaten society along the river route, also downstream of the extreme precipitation event. On the other hand, larger-scale floods in larger rivers with a slower increase of water levels are typically caused by longer-lasting heavy precipitation over a larger area. A multitude of atmospheric drivers can contribute to development of such extreme precipitation events and floods in different regions around the globe, such as convective cells, mesoscale convective systems, monsoonal lows or intense upper-level troughs or cut-off lows (Alcântara et al., 2023; Mohr et al., 2023; Ruff and Pfahl, 2023; Gaume et al., 2016; Gale and Saunders, 2013). In addition, orographic enhancement of precipitation can contribute to the development of floods in complex terrain.



The increase in global population, especially in high-risk areas, and the rise of the vulnerability of the society increases the risk of flood disasters (Kron, 2015). Moreover, climate simulations project that the frequency and intensity of extreme precipitation events is going to increase in a warmer climate (Pendergrass and Hartmann, 2014; Fischer et al., 2013; O'Gorman and Schneider, 2009), which is also associated with an increasing frequency of intense floods (see e.g. Alfieri et al., 2015). This, and a higher exposure of a growing proportion of the population to floods in a warmer climate, will drastically increase the flood risk on a global scale (Tellman et al., 2021; Alfieri et al., 2017; Jongman et al., 2012). In order to reduce or even prevent flood losses and damages, different kinds of flood protection measures have been developed. More recent extreme events have shown that flood damages have been reduced with the help of flood barrages compared to extreme precipitation that occurred several decades ago (see e.g. Merz et al., 2014; Bissolli et al., 2011), and that there is further potential to minimise flood losses in the future (Jongman et al., 2015). To this end, e.g., for the appropriate construction of dikes, it is crucial to precisely determine the precipitation amounts of potential extreme events that the protection measures should be able to withstand. In practical water management, such events are often denoted as "probable maximum precipitation" (Organization, 2009).

From a statistical point of view, this probable maximum precipitation can be quantified as the precipitation amount associated with extreme events with long return periods, typically on the order of 100 years, through extreme value statistics. To estimate such return values, long precipitation time series are required, which are conventionally obtained from observations. Although there are several observational data sets available with high quality and relatively high temporal and spatial resolution and coverage, this approach is affected by certain limitations (Rajulapati et al., 2020). First, the time series are often shorter than 100 years, which requires extrapolation to determine 100-year or larger return values and increases the associated statistical uncertainties. Second, if global coverage is desired, the uneven spatial distribution of rain gauges requires combining different data sources (e.g., rain gauge and satellite data), which can lead to spatial inhomogeneity in the estimated return values and associated uncertainties. Combining different data sets with diverging representations of extreme precipitation can also lead to inconsistencies on local scales (see e.g. Rajulapati et al., 2020). Third, precipitation trends, e.g., due to anthropogenic climate change, can compromise the extreme value statistics. Accordingly, Rajulapati et al. (2020) have shown that precipitation observations typically do not provide a consistent representation of extreme events and that 100-year return values differ significantly between observational data sets. Due to these limitations, previous studies often focused on extreme events with return periods of much less than 100 years (Rodrigues et al., 2020; Donat et al., 2013), for which observational time series are sufficient, and/or specific regions (Rodrigues et al., 2020; Maraun et al., 2011). Alternatively, model simulations, for instance from weather prediction (Ruff and Pfahl, 2023), seasonal forecasts (Kelder et al., 2020) or climate models (Mizuta and Endo, 2020) can provide long time series that allow for a statistically robust estimate also of 100-year return values. Nevertheless, these model-based estimates may of course suffer from other biases due to the imperfect representation of precipitation processes in the models.

In this study, we explore the possibility to use a model-generated data set from ensemble weather prediction for estimating 100-year return values of daily precipitation on a 1°x 1° grid covering the entire globe, extending our previous analysis that focused on European river catchments (Ruff and Pfahl, 2023). The equivalent length of the weather prediction data set is about 1200 years, that is much longer than the 100-year return period, promoting statistical robustness. Due to the daily accumulation,





relatively large spatial scale, and limited model resolution, the studied extreme precipitation events are most relevant for larger-scale river floods and not so much for local flash floods triggered by convective precipitation. The main goal of the study is to compare these model-based 100-year return values to estimates from three different observational data sets and quantify their relative biases and statistical uncertainties. This may provide a basis for using the model-generated data set also for practical
estimates of probable maximum precipitation, in particular in data-sparse regions.

The following Sec. 2 describes the ensemble weather prediction data as well as three observational data sets that are applied to evaluate the differences between the model-based and observation-based estimates of 100-year return values. In Sec. 3, the statistical methods to evaluate the ensemble prediction data and determine return values and confidence intervals are explained in detail. The resulting return values, their confidence intervals and differences to observational data sets are presented in Sec. 4.
Finally, conclusions and a discussion of the main findings and their limitations are provided in Sec. 5.

## 2    Data

100-year return values are estimated from a large global data set of daily precipitation events, which is obtained from ensemble weather prediction data. The resulting estimates are compared to observational data sets obtained from rain gauge measurements and satellites for evaluating the differences between a model-based and an observation approach. All data sets
are described in the following.

### 2.1    Ensemble prediction data

In order to generate a large data set of realistic and daily precipitation events, ensemble weather prediction data are used. In this study, the ensemble prediction system (EPS) of the European Center for Medium-Range Weather Forecasts (ECMWF) is accessed for this purpose. The ensemble predictions from the EPS are obtained from the Integrated Forecasting System (IFS),
which is a comprehensive earth system model with an atmospheric component from the ECMWF and other community models for certain other components of the earth system (ECMWF, 2023d). More details regarding the IFS and the operational EPS forecasts are described in ECMWF (2023c). An operational weather prediction model is very useful for such an approach as the model is capable of representing daily precipitation events more realistically than e. g. climate models due to a comprehensive comparison to observations (even without a surface precipitation assimilation scheme). Nevertheless, the EPS data may suffer
from inter-dependence between the ensemble members, which is investigated in more detail in Sec. 3.1, and from temporal inhomogeneities due to updates of the prediction system. The latter as well as additional limitations of the application of ensemble weather prediction data for the approach in this study are discussed in more detail in Sec. 5.

Started in March 2003, ensemble simulations of the operational weather prediction model are performed twice a day, at 0 and 12 UTC, with forecasting times of at least 10 days. The ensemble contains 51 ensemble members. One member is a
controlled run without any perturbations while the other 50 members represent runs with marginally changed initial conditions between each other and with stochastic perturbations of the model physics. This results in 102 simulations per real day. More information about the workflow of the EPS can be found in Molteni et al. (1996).





The analyses in this study are based on daily precipitation sums, which are computed by adding up the large-scale and convective precipitation over 24 hours. From each simulation, the daily precipitation sum just of the 10th forecast day (between

forecast hours 216 and 240, same procedure for every initialisation time) is selected, instead of using all forecast days. This approach follows Ruff and Pfahl (2023), who used a daily precipitation data set from the EPS to investigate the atmospheric conditions during extreme precipitation events over Central Europe, and Breivik et al. (2013), who estimated return values of oceanic surface wave heights from EPS data. The basis of the approach is the assumption that, due to the advanced forecast time, the model realisation on the 10th forecast day do not significantly correlate with the conditions in the beginning of each

specific simulation. Therefore, the different realisations obtained from the ensemble members can be considered as statistically independent from each other. While the simulations of individual ensemble members are highly correlated to each other in the beginning due to very similar initial conditions, this correlation reduces with increasing forecast time. This decrease is particularly large for precipitation, compared to, e.g., geopotential height, due to its high variability in space and time and its dependence on small-scale processes. Both Ruff and Pfahl (2023) and Breivik et al. (2013) performed comprehensive statistical

analyses to demonstrate the independence of the ensemble members on the 10th forecast day and to compare the statistics of daily precipitation and wave height to observational data. The data set used here is very similar to the data of Ruff and Pfahl (2023), except that they analysed spatially averaged precipitation time series over Central European river catchments, while this study uses time series on a spatial grid of $1° \times 1°$ spanning the entire globe. Therefore, just a short statistical evaluation of the ensemble prediction data (see Sec. 3.1), adapted to time series at individual grid point, is performed in this study, while we

refer to Ruff and Pfahl (2023) for other, more detailed statistical analyses.

The operational model IFS has been updated on a regular basis. Certain technical and physical schemes were changed with each implementation of a new model cycle during the years 2003-2019 in order to continuously improve the forecast skill. However, this may lead to the potential of influencing the simulated precipitation and the upcoming results of this study. Although mainly minor improvements were implemented within each individual model cycle, there are some important updates

that include a changed formulation of the humidity analysis (Cycle 26r1), improved precipitation forecasts over Europe (Cycle 32r3) and improved precipitation forecasts over coastal areas due to changes in cloud physics (Cycle 45r1). Details of all model cycle changes are described in ECMWF (2023a) and a full documentation of each model cycle itself is available from ECMWF (2023c). Ruff and Pfahl (2023) have evaluated the influence of these model cycle updates on their daily precipitation time series. They demonstrated a systematic decrease of high precipitation percentiles (99th, 99.9th and 99.99th), which corresponds

to extreme precipitation events with large return periods, over Central Europe within the first five years of the ensemble simulations (2003-2007). On the contrary, since 2008, the amplitude of these percentiles is rather constant, as discussed in more detail in Sec. 3.1. Hence, in order to avoid any temporal inconsistencies within the data set, only the ensemble simulations from the 1st of January 2008 until the 31st of December 2019 are used in this study, following Ruff and Pfahl (2023). The restricted time period of 12 years of forecasts from the EPS archive along with 102 daily simulations, provides a data set

with an equivalent length of 1224 years of modelled, but realistic daily precipitation events. The data set is available on a regular lat/lon grid with varying resolutions over time due to changes in the forecast model cycles. For a consistent analysis



and comparison to observational data, they are re-gridded to a uniform resolution of $1°\text{x } 1°$ with the ECMWF interpolation scheme MIR (ECMWF, 2023e).

## 2.2 Observational data sets

One observational data set based on rain gauge measurements (REGEN) and two data sets based on a combination of satellite data and rain gauges (CHIRPS and PERSIANN) are used to compare daily 100-year precipitation return values and their confidence intervals from the EPS forecasts to observations. The observational data sets mainly differ in their covered region, the type and amount of observations and the interpolation of the data to a regular lat/lon grid. All observations are described in the following.

### 2.2.1 REGEN data


The Rainfall Estimates on a Gridded Network (REGEN) is an observational data set for daily precipitation which uses quality controlled rain gauge measurements, spatially interpolated from daily precipitation data of large observational archives such as the Global Historical Climate Network Daily, provided by the National Oceanic and Atmospheric Administration, and the Global Precipitation Climatology Centre, provided by the Deutscher Wetterdienst. More information about this data set are
presented in Contractor et al. (2020a). The spatial density of available rain gauges is very different between certain regions. Especially over Africa and Central Asia, the rain gauge density is considerably lower than over North America, Europe and Australia. While this study only uses the daily precipitation sums (Version 1-2019), other information are additionally available on a global grid such as e. g. the number of rain gauges and the standard deviation of the precipitation sums. The precipitation data are constructed from around 135.000 rain gauges between the 1st of January 1950 and the 31th of December 2016 while
not all stations are available for the entire period and can be used on a regular lat/lon grid with a spatial resolution of $1°\text{x } 1°$ for all global land areas, except for Antarctica.

### 2.2.2 CHIRPS data

The Climate Hazards Group Infrared Precipitation with Stations (CHIRPS) data archive is a quasi-global, daily precipitation data set, hosted by the U.S. Geological Survey Earth Resources Observation and Science Center in collaboration with the Santa
Barbara Climate Hazards Group at the University of California. The CHIRPS data result from a combination of quasi-global geostationary thermal infrared satellite observations from two NOAA sources, in situ precipitation observations obtained from a variety of national and regional meteorological services, the Tropical Rainfall Measuring Mission product from NASA, a monthly precipitation climatology and atmospheric model rainfall fields from NOAA. A more detailed description of the development workflow of the CHIRPS data can be found in Funk et al. (2014a). The data are available from the 1st of January
1981 until the 31th of December 2021 over land areas on a regular lat/lon grid between 50°S and 50°N. In this study, the CHIRPS data (Version 2.0) with a spatial resolution of $0.25°\text{x } 0.25°$ are selected and interpolated (averaged over several $0.25°$ boxes) to a $1°\text{x } 1°$ grid for further analyses.





### 2.2.3 PERSIANN data

The Precipitation Estimation from Remotely Sensed Information using Artificial Neural Networks–Climate Data Record
(PERSIANN-CDR), hosted by the Center for Hydrometeorology and Remote Sensing at the University of California, are
satellite based, daily precipitation observations. This data sets results from gridded satellite infrared data, obtained from a combination of several international geostationary satellites, in combination with an artificial neural network training using hourly
precipitation data from the National Centers for Environmental Prediction stage IV. Additionally, the monthly product of the
Global Precipitation Climatology Project is used for bias adjustments. Further details on this data set are described in Ashouri
et al. (2015a). The daily precipitation sums of this data set (Version 1) are available from the 1st of January 1983 until the 31th
of December 2021 on a regular lat/lon grid between 60°S and 60°N with a spatial resolution of 0.25°x 0.25°. In this study, the
data are interpolated to a 1°x 1° grid for further analyses. Missing values in the data set appear in cases when satellite data are
not available and on dry days when no precipitation occurred (see Supplementary Fig. S1). As an under-representation of dry
days strongly influences the quantile distributions of the data (used for evaluations in Sec. 3.1), all missing values are set to 0
for further analyses, in order to improve the representation of the percentiles. However, this also leads to errors over areas that
are regularly affected by non-availability of satellite data. Especially at around 50°N and 70°E as well as 50°S and 70°E this
is most dominant, which is why these areas should be taken into account when interpreting the results from the PERSIANN
observations.

## 3 Methodology

In this section, statistical analyses of the suitability of the EPS data for determining 100-year precipitation return values on a
global grid are presented. Subsequently, the method to determine the return values and their confidence intervals is described.

### 3.1 Statistical evaluation of the ensemble prediction data

For the determination of 100-year return values of daily precipitation on a global scale, the EPS data at the 10th forecast day
are used as a large climatological data set. This implies that the data set can be considered as a combination of realistic and
independent realisations of daily precipitation in order to suitably apply extreme value statistics to this data set. Therefore,
we evaluate if (1) the ensemble members can be considered as independent from each other, (2) each ensemble member
properly represents the statistical distribution of precipitation compared to observations and (3) no significant trend of the high
precipitation percentiles can be identified over time, following Ruff and Pfahl (2023) and Breivik et al. (2013). These criteria
are statistically evaluated and discussed in the following based on time series of daily precipitation sums on the 10th day of
each forecast and at each global grid point. This spatial coverage is the main difference between this study and Ruff and Pfahl
(2023), who analysed time series of spatially averaged precipitation in several Central European river catchments.

Beginning with the first criterion, the daily precipitation at the 10th forecast day of each individual ensemble member is
investigated with regard to its independence to each other. For this, Ruff and Pfahl (2023) analysed the statistical distribution of



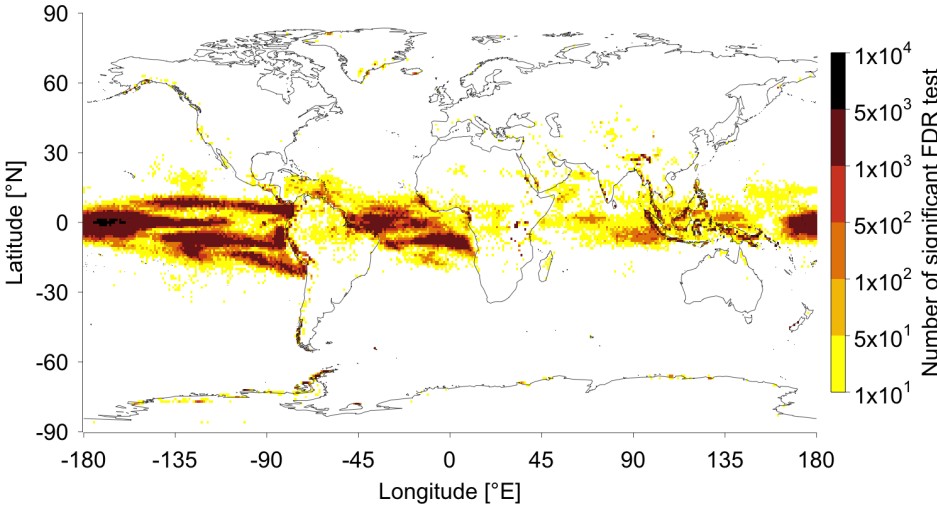

**Figure 1.** Number of statistical significant correlation coefficients per grid point, obtained from the FDR tests of Benjamini and Hochberg (1995), as described in Ventura et al. (2004), applied to multiple (5152) $p$ values associated with the Pearson correlations between yearly maxima time series of individual ensemble members. Mind the logarithmic colour scale.

Spearman correlation coefficients between all possible ensemble member combinations (5151 in total) for time series of both all daily precipitation sums and their annual maxima (which are used to determine the 100-year return values, see Sec. 3.2). They showed that the time series based on all daily events are weakly, but still significantly, correlated between the ensemble members (mean correlation around 0.19, similar for all river catchments). Time series of yearly maxima of daily precipitation sums are generally very weakly correlated as well (mean around 0), but there is almost no significant correlation identified, supporting the assumption that daily extreme precipitation is independent between ensemble members. Here, this correlation analysis is expanded to the global grid, leading to an analysis of multiple correlations. In order to determine the statistical significance of the correlation coefficients in such a multi-test framework, the False Discovery Rate (FDR) test of Benjamini and Hochberg (1995), as described in Ventura et al. (2004), is applied to all p-values of the correlation coefficients from each combination of ensemble members at a certain grid points. Figure 1 shows the number of statistically significant correlation coefficients from the FDR test for time series of annual maximum daily precipitation at each global grid point. For most of the grid points, (almost) no significant correlations are found, especially poleward of 20°N and 20°S, supporting the hypothesis that annual maximum precipitation events are independent between ensemble members on a global scale. However, there are certain areas over the tropical oceans, the Maritime Continent and South-East Asia, where FDR tests show high numbers of significant correlation coefficients. In these regions, extreme precipitation events in the individual ensemble members show a certain dependence, and the EPS data set cannot be considered as equivalent to a time series of 1224 years in the analysis of 100-year return values. A likely reason for this inter-dependence is the influence of internal climate modes with relatively long time scales, such as the El Niño-Southern Oscillation, on tropical precipitation events, which can lead to a synchronisation of annual maxima between ensemble members.



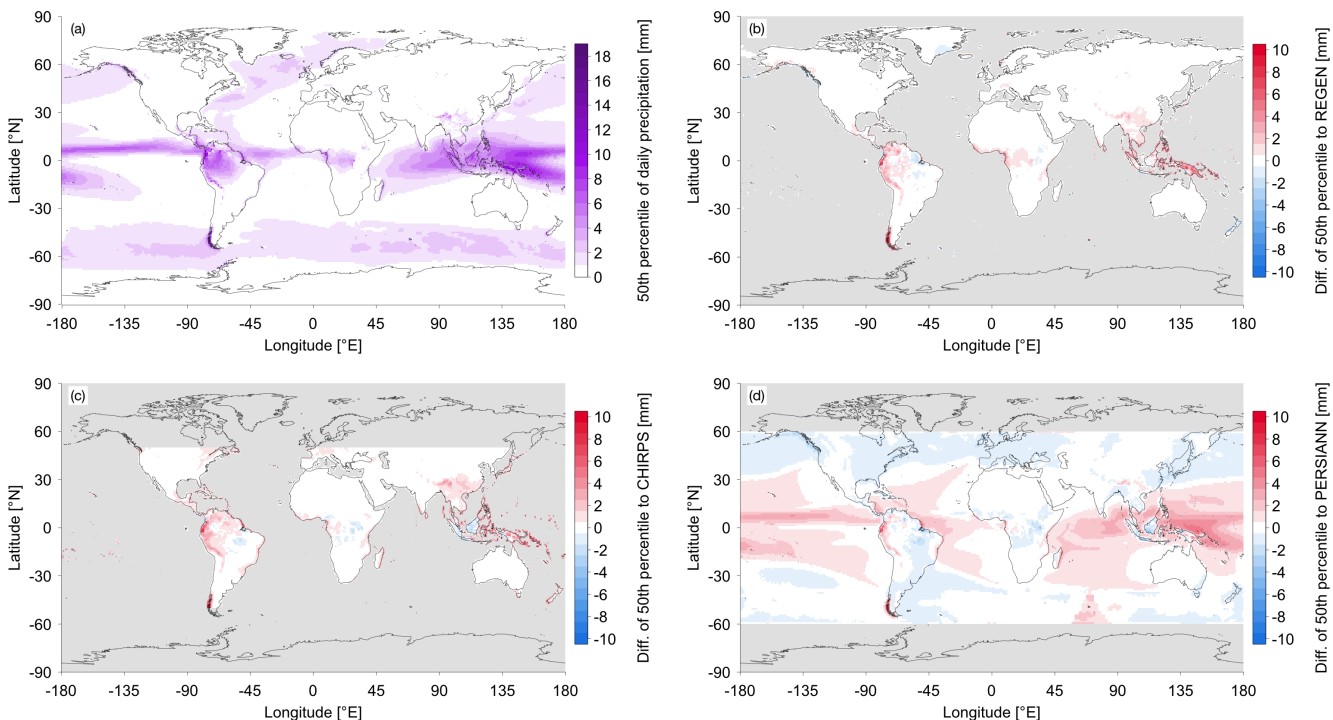

**Figure 2.** Global distribution of the 50th percentile of daily precipitation from **(a)** the EPS data and their difference to the observational data sets **(b)** REGEN, **(c)** CHIRPS and **(d)** PERSIANN. Areas where the observational data are not available are shown in light grey.

To evaluate the statistical distribution of daily precipitation, Ruff and Pfahl (2023) compared quantiles from the EPS data to three observational data sets (REGEN, ENSEMBLES daily gridded observational dataset for precipitation, temperature and sea level pressure in Europe (E-OBS) and Hydrometeorologische Rasterdaten (HYRAS)) and found a good agreement in the Central European river catchments. For a similar analysis on the global scale, we select the 50th and 90th percentiles at each grid point. Note that all days of a time series, including dry days, are considered for determining these percentiles (Pfleiderer et al., 2019). Figure 2 shows the 50th percentile (median) from the EPS data (taking all members together) in (a) and the differences to the observational data in (b-d). Higher median precipitation amounts can generally be found over the oceans while the highest values lie within the tropics, in the area of the Intertropical Convergence Zone (ITCZ). Over the extratropical continents, due to a relatively high number of dry days, the median of daily precipitation is mostly below 1 mm. The differences of the 50th percentile to the REGEN and CHIRPS data set indicates a relatively good agreement over most continental regions (see Fig. 2b,c). Larger differences are obtained in the tropics, in particular over the northern part of South America, the Maritime Continent and part of South-East Asia, where the 50th percentile of the EPS data is up to 8 mm higher than for both observational data sets. The differences to the PERSIANN observations show a similar pattern over continental areas (see Fig. 2d), however, also revealing that the 50th percentile of the PERSIANN data is slightly higher (by 1 mm) over North and South America and Europe compared to the EPS data. Larger differences are found over the tropical and extratropical



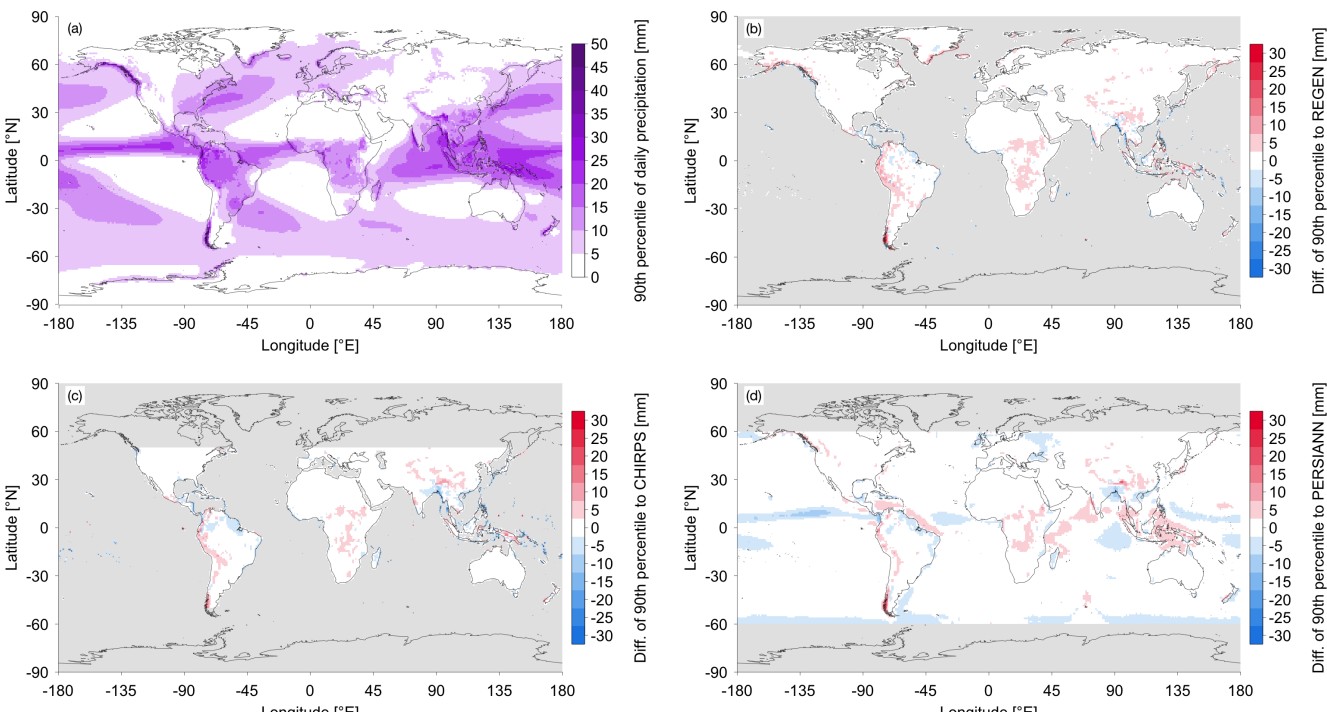

**Figure 3.** As Fig. 2, but for the 90th percentile.

oceans with a higher median in the EPS data by 1 to 3 mm and up to 5 mm over the tropical West Pacific. Similar results are obtained for the 90th percentile (see Fig. 3). Most of the differences to the REGEN and CHIRPS observations (see Fig. 3b,c)
are close to 0, however, larger differences are found over Africa, the west coast of South America (overestimation of up to 10 mm) and South-East Asia (under- and overestimation of up to 20 mm). The differences to the PERSIANN data show a very similar pattern over the continental areas (see Fig. 3d). Additionally, the differences over the oceans are also very small, except some minor over- and underestimations in the tropics by around 5 mm. In summary, the EPS data realistically represent daily precipitation statistics in most regions of the globe, with some larger biases over the tropics and a few high-altitude regions,
e.g., in South America. Ruff and Pfahl (2023) also compared the precipitation statistics between each individual ensemble member, but as they found no systematic differences, this analysis is not repeated here.

Stationarity of the time series is important for applying the extreme value analysis described in Sec. 3.2. To evaluate the stationarity of the EPS data, Ruff and Pfahl (2023) applied the Mann-Kendall test to several high percentiles and the yearly number of 100-year precipitation events for the years 2008 to 2019 and also compared the occurrences of such events to
a Poisson distribution of independent events with a constant mean rate. They did not find an indication of temporal non-stationarity in this period of 12 years. Here, the same Mann-Kendall test is applied to the 99.9th percentile of daily precipitation (representing an almost 3-year precipitation event) at all global grid points (see Supplementary Fig. S2). In order to evaluate the statistical significance of these multiple tests, again the FDR test is applied as introduced earlier. No significant trend is





found for the EPS data. For the observational data sets, also due to the longer time series covering 39 years or more, some

areas are associated with significant trends, rather evenly distributed over the globe (for REGEN, 8.4% of grid points with significant trends), over Central Africa and Central Asia (for CHIRPS, 2.4% of grid points with significant trends) or India and the Indian Ocean (for PERSIANN, 3.4% of grid points with significant trends). Nevertheless, trends are still not significant over the most part of the globe. In order to use a consistent methodology for all data sets and locations, we thus make the stationarity assumption also for the observational time series.

In summary, our analyses show that, in most regions, daily precipitation obtained from different members of the ECMWF EPS can be considered as statistically independent. Exceptions are some areas over the tropical oceans and Maritime continent (see again Fig. 1). Additionally, the model data realistically represent different quantiles of daily precipitation in comparison to three observational data (see Figs. 2 and 3), again with the exception of a few regions mostly in the tropics. Finally, there is no indication of non-stationarity in the data set over the time frame analysed here. Thus, we consider the EPS data to be suitable

for a global analysis of 100-year return values of daily precipitation.

## 3.2 Determination of return values and confidence intervals

To determine 100-year return values of daily precipitation and their confidence intervals at each grid point, the daily precipitation sums from all ensemble members are used to build long time series. For this investigation, extreme value statistics (see Coles et al., 2001) are applied in order to fit a generalised extreme value distribution (GEV) to a selected sample of block

maxima, using the maximum likelihood approach. This sample of block maxima is here selected from yearly blocks of daily precipitation. Such an approach lead to 1224 block maxima at each grid point. This number of block maxima is sufficient for the Fisher-Tippett theorem, so that the GEV can be fitted to the selected block maxima. The best fit of the GEV is accomplished by estimating the location ($\mu$), scale ($\sigma$) and shape ($\xi$) parameters. Following Stephenson (2002), the return value $v$ can be computed from these estimated parameters by the following equation:

$$
\begin{aligned}
v &= \mu + \sigma \cdot \frac{(x^\xi - 1)}{\xi} \\
x &= \frac{-1}{log(1 - \frac{1}{p})}
\end{aligned}
\tag{1}
$$

in which a certain yearly return period in described by $p$. The confidence intervals of the return values are computed by the bootstrap resampling method (see Coles et al., 2001). Taking the original set of block maxima, a new set of maxima is drawn with replacement. Then, the previously explained approach of fitting a GEV to a selected sample of block maxima is repeated

with the new set of block maxima and the return value is again determined from Eq. (1). As each procedure leads to a slightly changed return value, the uncertainty can be evaluated by repeating this process several times. Here, it is repeated 1000 times. From the resulting 1000 return values, the 0.025 and 0.975 quantiles are considered to be the confidence intervals of the return value from the original sample of block maxima.



At some grid points with very low precipitation amounts during the entire year (e.g., over the Sahara Desert), the best fit
of the GEV yields very high estimates of the shape parameter $\xi$ (up to 3). This results in extraordinary high return values
compared to neighbouring grid points with low return values. Papalexiou and Koutsoyiannis (2013) analysed estimated shape
parameters from GEV fits for over 15.000 globally distributed observational records, using yearly maxima of daily precipitation
as block maxima as well. Even for rather short time series of at most 10 years, which are often associated with higher shape
parameters than longer time series, the shape parameters all lie between -0.6 and 0.6, independent of their location. However,
almost no time series of their observational data set are located in very dry regions such as the Sahara Desert. In order to
prevent unrealistic high or low return values but still allow shape parameters outside the range of -0.6 and 0.6 for areas that are
typically not covered by observations, grid points with estimated shape parameters above 1 or below -1 and scale parameters
above 70 or below -70 are excluded from the analyses in this study.

## 4 Results

In this section, the estimated 100-year return values of daily precipitation and their confidence intervals are presented. Estimates
from the EPS data are compared to the observational estimates based on REGEN, CHIRPS and PERSIANN.

### 4.1 Return values

Global estimates of 100-year return values of daily precipitation from the EPS data set are shown in Fig. 4a. Generally, higher
return values can be found in tropics and parts of the subtropics, while return values decrease towards the poles. The return
values range from 0.98 mm over Antarctica up to 985.36 mm over the Arabian Sea, near the southeast coast of the Arabian
Peninsula. The area of the Arabian Sea generally shows the highest return values, ranging from 600 mm to 900 mm, followed
by parts of India, the Himalayas, South-East Asia and the ITCZ regions of the Atlantic and Pacific Ocean with return values
of 400 mm to 600 mm. Return values of about 300 mm are found for most other parts of the tropical and subtropical oceans,
the northern part of South America, the region south of the Sahel and northern parts of Australia. North America, Europe and
northern Asia are associated with return values of around 100 mm, while a 100-year event over most regions poleward of 60°N
and 60°S does not exceed 50 mm. Typical dry regions such as the Sahara Desert and the Tibetan Plateau are also associated
with very low 100-year return values.

A 100-year daily precipitation event over the southeast coast of the Arabian Peninsula would exceed the local annual mean
precipitation by a factor of seven or higher (see Fig. 5). Very high exceedances of the annual mean can also be found over
certain oceanic regions near the west coasts of Africa, North and South America and over regions with low return values such
as the Sahara Desert and the Taklamakan Desert. A 100-year extreme event over Australia would contribute about 20% to
60% (depending on the region) to the annual mean, while this percentage typically ranges around 10% for most of the other
continental areas such as North and South America, Europe, Central and South Africa and Asia.

A comparison with the observational data set REGEN shows generally higher EPS estimates over large parts of the globe
and very large differences in several tropical and subtropical regions (Fig. 4b). The 100-year return values from the EPS data



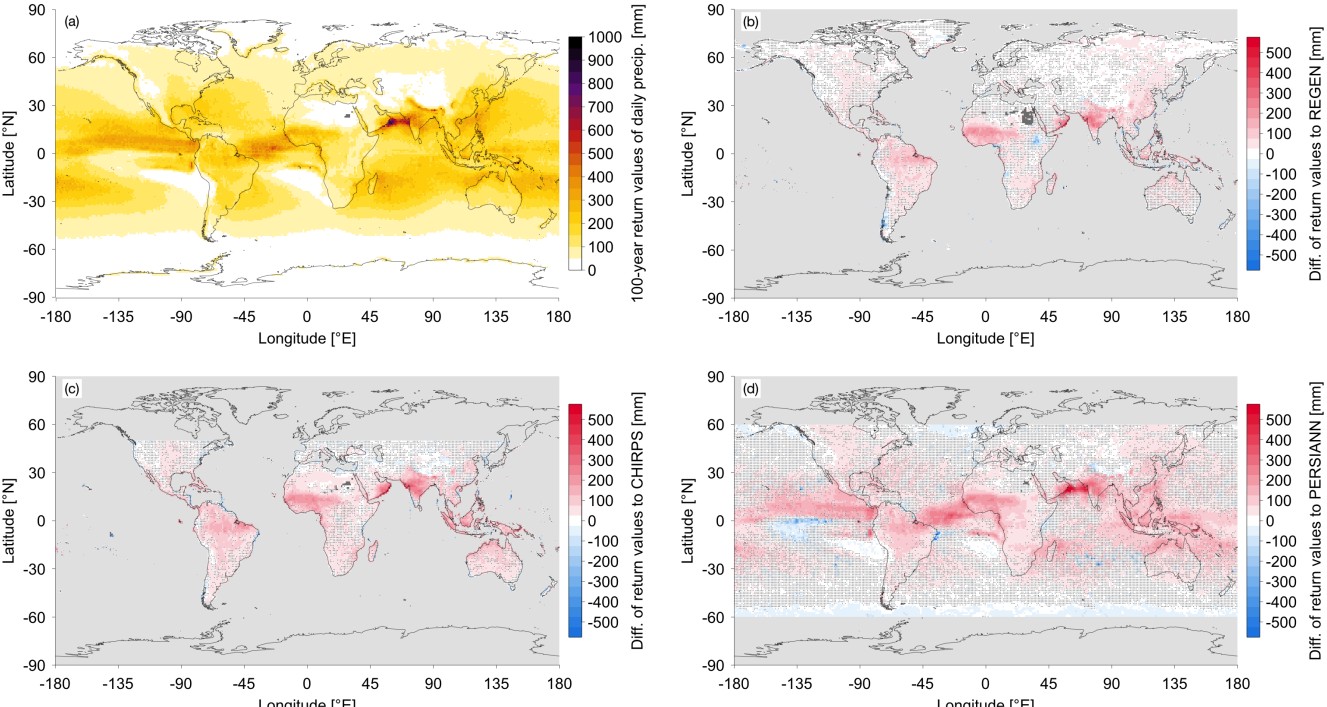

**Figure 4. (a)** Global distribution of 100-year return values of daily precipitation, estimated from EPS data, and their difference to the observational estimates from **(b)** REGEN, **(c)** CHIRPS and **(d)** PERSIANN. Areas where the observational data are not available are shown in light grey. Dark grey shading indicates grid points for which the GEV parameters are outside the allowed range and thus no return values can be estimated. Stippling in **(b)**-**(d)** shows where the confidence interval of the EPS data overlaps with the confidence interval from the specific observational data set.

are clearly higher over the southeast coast of the Arabian Peninsula (by 300-500 mm), India (by 100-400 mm), western Africa (by 100-300 mm) and over the Amazon (by 100-200 mm). Additionally, the confidence intervals of the EPS data do not overlap with the REGEN data in these regions, the differences are thus statistically significant. Over other parts of South America, the southern half of Africa, South-East Asia and Australia, the EPS estimates are about 50-100 mm higher, but the
confidence intervals overlap in parts of these areas. Most other regions do not show large differences in the estimated return values. However, especially over parts of Chile, the Abyssinia Plateau in East Africa and over some coastal areas in South-East Asia, the EPS data are associated with lower return values than the REGEN data (no overlap of confidence intervals). Overall, the confidence intervals of EPS and REGEN return value estimates overlap at 50.4% of the grid boxes. The differences of EPS return values to the CHIRPS and PERSIANN observational estimates (Figs. 4c,d) are very similar to each other. The
confidence intervals of EPS and CHIRPS overlap at 45.1% of the grid boxes, and for EPS and PERSIANN this overlap is 55.9%. Furthermore, the CHIRPS and PERSIANN results are mostly consistent with the differences to the REGEN estimates described above. However, the strongly negative differences over Chile and the Abyssinia Plateau do not occur for the CHIRPS



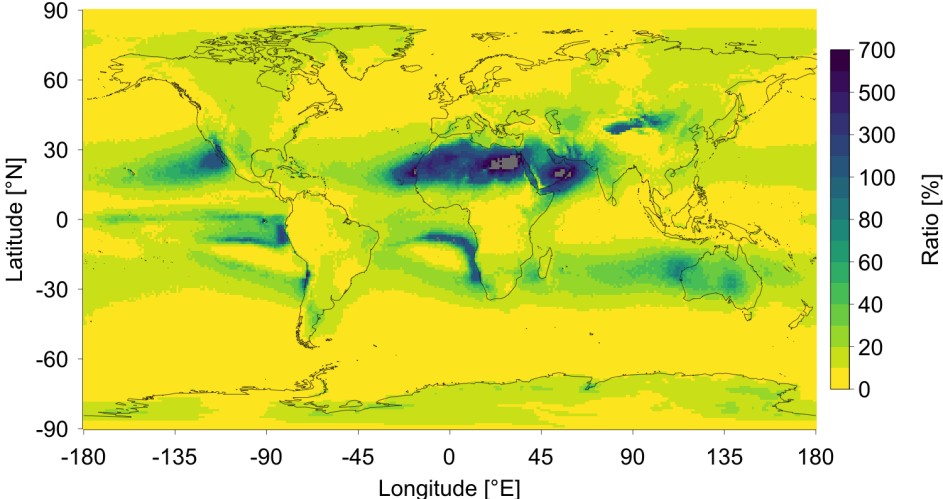

**Figure 5.** Ratio between the precipitation amount of a 100-year event of daily precipitation and the annual mean precipitation from the EPS data in %. The annual mean precipitation is averaged over all ensemble members and over the years 2008 to 2019. Mind the non-linear colour scale. Values above 700% are shown in grey.

and PERSIANN data. Additionally, the positive differences over the Arabian Peninsula, India, South-East Asia and Australia are even larger. In the comparison with PERSIANN, also oceanic regions can be considered. In particular in the ITCZ region,
around Madagascar and over the eastern Indian and western Pacific Ocean, the EPS data produce higher return values than PERSIANN. There are also some negative differences over the tropical Pacific, the most eastern tip of the Brazilian coast and for most areas poleward of 50°N and 50°S. It should be kept in mind here that, over parts of the tropical oceans, the EPS estimates are less trustworthy than in other regions due to methodological issues associated with inter-dependence between the ensemble members (see again Sec. 3.1).

**4.2  Confidence intervals**

Confidence intervals (CIs) on a 95% level are determined for each return value estimate. To compare the confidence interval ranges (that is, the difference between upper and lower bounds) between data sets, on the one hand, in Fig. 6 they are shown relative to the corresponding return value estimate, as higher return values are typically associated with larger confidence intervals. In this way, the relative uncertainty of the 100-year return values is quantified. On the other hand, Fig. 7 shows the
absolute confidence interval ranges, quantifying the absolute uncertainties.

The largest relative uncertainties are found in tropical and subtropical regions. Especially over the Sahara Desert, the south-eastern part of the Arabian Peninsula and some areas over the tropical Pacific and Atlantic Ocean, the relative CIs range from 30% to 80% (see Fig. 6a). However, there is no clear pattern that higher CI ranges predominantly occur over regions with particularly high or low return values. For instance, high relative uncertainties are found in the dry Sahara Desert, but also
over the wet Arabian Sea. Over most other parts of the tropics and subtropics, the range of the CIs lies between 10% and





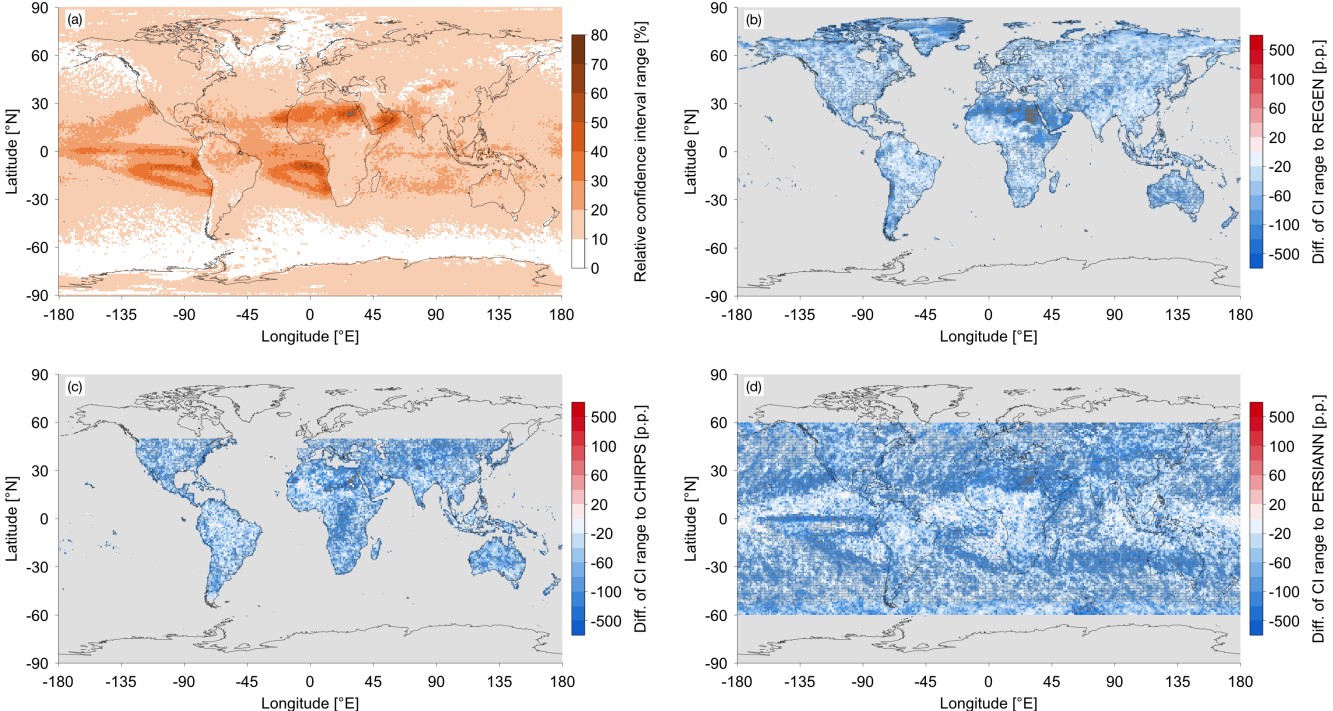

**Figure 6. (a)** Global distribution of the relative range of confidence intervals on a 95% level, relative to the associated return values, and their differences to the observational estimates from **(b)** REGEN, **(c)** CHIRPS and **(d)** PERSIANN. Areas where the observational data are not available are shown in light grey. Dark grey shading indicates grid points for which the GEV parameters are outside the allowed range and thus no return values can be estimated. Stippling in **(b)**-**(d)** shows where the confidence interval of the EPS data overlaps with the confidence interval from the specific observational data set. Mind the non-linear colour scales. In **(b)**-**(d)**, only values between -700 p.p. and 700 p.p. are displayed.

30%. Poleward of 30°N and 30°S, the relative CI ranges are typically around 10%. As mentioned before, higher return values are associated with higher CIs, as shown by the absolute values in Fig. 7a. The tropics and several subtropical areas have the highest absolute CI ranges, especially over the Arabian Sea (up to 400 mm) and the tropical Pacific and Atlantic (100 mm to 300 mm). Absolute CIs over North America, Europe and northern Asia, typically lie around 10 mm.

The differences of the relative CI ranges between the EPS and REGEN data sets are shown in Fig. 6b. Over almost all continental areas, the relative uncertainties are reduced in the EPS data compared to CHIRPS, with typical differences on the order of 50-100 p.p.. Over specific regions such as the west coast of South America, the Sahara Desert, the Arabian Peninsula and Greenland, this decrease is even larger (-200 p.p. to -800 p.p.), while the CIs mainly overlap with each other in these regions. A slight increase in the relative uncertainty by around 10 p.p. is found for some grid points over the Amazon, West
Africa, India and China. In terms of absolute CI ranges (Fig. 7b), the uncertainty is reduced in the EPS data compared to REGEN by up to 300 mm over South-East Asia and Australia and around 50 mm over North America and Europe. Note that





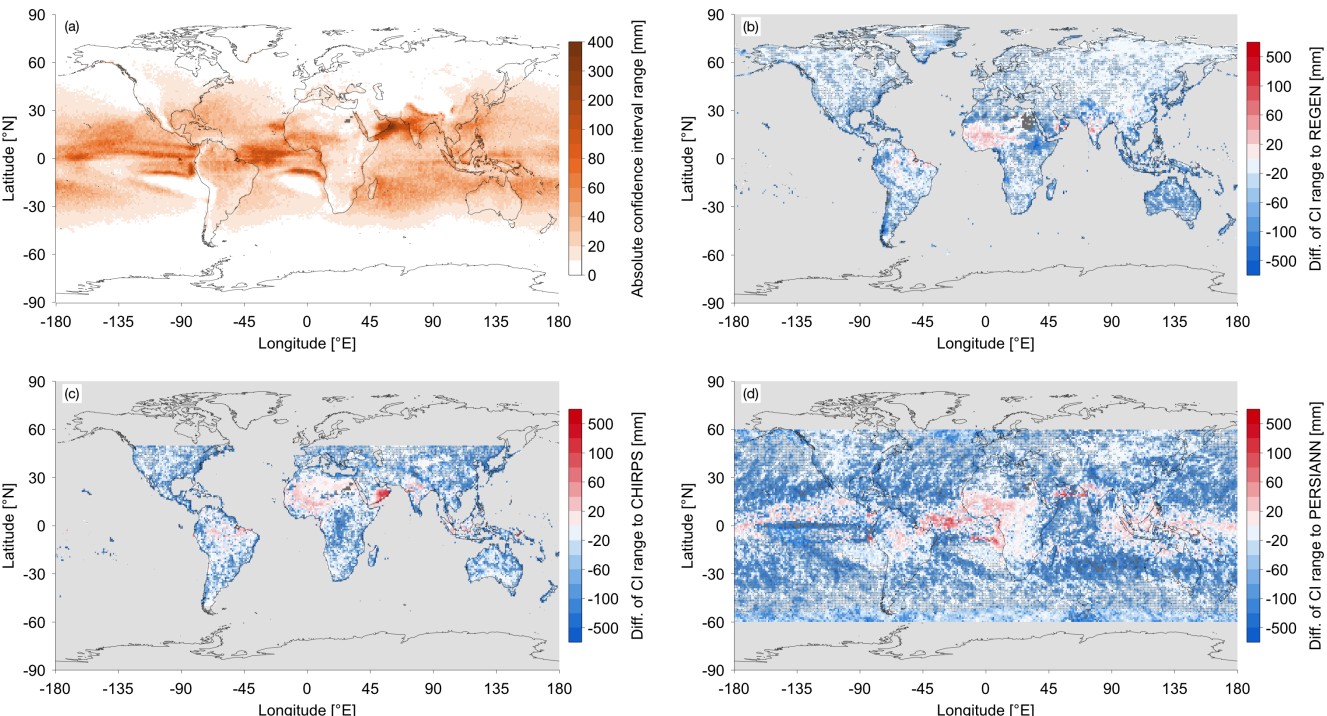

**Figure 7.** As Fig. 6, but for the absolute range of confidence intervals in mm.

this corresponds to a reduction by at least a factor of two, that is a substantially reduced uncertainty, in most regions. There are a few more extended (compared to the relative CI ranges) areas of larger absolute uncertainties in the EPS data over the Amazon, West Africa and India with increases of 20 mm to 100 mm. These are also regions where the CIs do not overlap.

Similar results are found for both relative and absolute uncertainties taking the CHIRPS data for comparison. The relative CI range is smaller in the EPS data over almost all regions (Fig. 6c), and the magnitude of this reduction is typically even larger (around -100 p.p.) than for REGEN, except for the Sahara Desert. Differences in the absolute uncertainty compared to CHIRPS are again similar as for REGEN, but with a more enhanced and broader increase in the southeast of the Arabian Peninsula of up to 400 mm (Fig. 7c). Finally, the uncertainty in EPS is also reduced compared to PERSIANN, with similar

patterns over the continents as for REGEN and CHIRPS (Fig. 6d). In addition, a reduction of relative CI ranges with a similar magnitude (around -100 p.p.) is also found over most oceanic regions, where CIs still overlap in the extratropics, but less so in the tropics. Some of these tropical regions with significant differences between the EPS and PERSIANN estimates of 100-year return values are associated with a larger absolute uncertainty in the EPS data by around 50 mm and up to 300 mm over the tropical Atlantic (Fig. 7d). Note that such a co-occurring increase of absolute uncertainty and decrease in relative uncertainty

is linked to substantially higher return value estimates in the EPS data.





## 5   Discussion and conclusion

The aim of this study has been to determine 100-year return values of daily precipitation and their confidence intervals (CIs) on a global scale from a large data set of model-generated events and to evaluate the differences to three observational data sets (REGEN, CHIRPS and PERSIANN). A quantification of such extreme return values is crucial for properly setting up flood

protection measures and, hence, reduce flood risks, also because such extreme events are expected to occur more frequently in a warmer climate. The large set of simulated global daily precipitation fields has been obtained from operational ensemble weather prediction data by the ECMWF, following the approach of Breivik et al. (2013) and Ruff and Pfahl (2023). Our statistical analyses show that, when using the 10th forecast days from these simulations, annual precipitation maxima are independent between the different ensemble members in most regions, except for the tropical oceans and the Maritime Continent, where

the results are thus less trustworthy. Biases in the climatological distribution of precipitation with respect to the observational data sets, evaluated through the 50th and 90th percentile, are also relatively small in most regions. Additionally, there is no significant trend in the occurrence of intense precipitation over the 12 years at all grid points, and the EPS data can thus be considered as stationary in time. With the help of extreme value statistics, 100-year return values and the associated confidence intervals on a 95% level are determined and compared to estimates from the observational data sets.

Based on these EPS data, the largest 100-year return values of daily precipitation occur in the tropics with a maximum around 950 mm (absolute CI range of 400 mm) over the Arabian Sea near the coast of the Arabian Peninsula. The return values decrease towards the poles with a minimum only slightly above 0 mm over parts of Antarctica. Over India, the Himalayas and the tropical Pacific and Atlantic, 100-year events are associated with precipitation amounts around 500 mm (absolute CI range around 200 mm), while for Australia, Africa, South America and the subtropical oceans they typically amount to about 200 mm

(absolute CI range around 50 mm). The return values over North America, Europe and Central Asia lie mostly between 50 and 100 mm (absolute CI range of 10 mm). Over continental areas, a 100-year event would typically amount to about 10-20% of annual mean precipitation (20-50% for Australia), but in dry regions such as the Sahara Desert and Arabian Peninsula the estimated 100-year return value even exceed the annual mean precipitation by a factor of up to 7. The 100-year return values from the EPS data are in general agreement with other studies of multi-year precipitation extremes. For instance, Rodrigues

et al. (2020) determined 10-year return values for Brazil and found slightly lower values and a local maximum of 200 mm further towards the east coast. Gründemann et al. (2023) studied 100-year return values over global land areas based on several statistical approaches and a data set obtained from a combination of satellite observations, reanalyses and gauge data. They found similar spatial patterns as documented here, but lower return values over, for instance, the Sahel, the east coast of the Arabian Peninsula and parts of India. Also the comparison with the observational data sets in this study indicates systematically

higher return values in the EPS data set over most of the globe. In many regions, in particular in the extratropics, the confidence intervals of EPS and observational estimates still overlap. Larger, also statistically significant differences are obtained in some tropical regions where the EPS method is less robust due to inter-dependence of ensemble members and/or general biases in the precipitation climatology, but also over other areas where the data set performs well in our statistical evaluation, such as parts of northeastern South America, western tropical Africa, India and eastern China. This might be due to model deficiencies,



e.g., in the representation of convective precipitation in the tropics, but may also point to a systematic underestimation of very extreme daily precipitation events in observations due to the relatively short time series and thus limited sampling. Such a potential under-estimation might have important consequences for practical applications and should be considered in estimates of "probable maximum precipitation" for designing flood protection measures.

The relative uncertainty of the 100-year return values is quantified through relative confidence interval ranges, which typ-
ically lie within 10-30% for most of the regions, but can be higher than 50% over the Sahara Desert, the Arabian Peninsula and parts of the tropical and subtropical Pacific and Atlantic. An important result is that these relative uncertainties, and in most regions also the absolute CI ranges, are substantially reduced in the EPS data compared to observational estimates. This reduction is typically on the order of 50-100 p.p., but can locally amount to up to 600 p.p., e.g., over the Sahara Desert and the Arabian Peninsula. The systematic and substantial reduction of statistical uncertainty is due to the much longer time series and
is the main advantage of the model-generated EPS data.

Our approach to estimate 100-year precipitation return values from EPS data has several limitations, as also discussed by Ruff and Pfahl (2023). First, the model-generated precipitation data may be affected by model biases and the imperfect representation of specific processes in the model. Such biases are assumed to be particularly large for small-scale, convective precipitation events, which is one reason why we focus on larger-scale events on a 1°x 1° grid. Second, the time span of the
forecast data is limited to 12 years (2008-2019) which comes along with a limited sampling of large-scale boundary conditions. Therefore, not the entire range of (multi-)decadal variability of the climate system is reproduced in the data set. Third, there is a certain influence of anthropogenic forcing in the data set in specific regions, as mentioned in Sec. 3.1, which can lead to temporal inhomogeneities. However, our trend analyses shows that this is mainly restricted to a few areas over tropical and subtropical oceans. Fourth, the method does not work well in regions where the ensemble members are not independent from
each other, which is mainly the tropical oceans and the Maritime Continent.

In future research, the approach used here may be applied for events with different return periods and also to other weather prediction ensemble data set, which may help in clarifying the reasons for the systematically higher return values in the EPS data compared to observations. Finally, large initial condition ensemble simulations with climate models may be used to investigate the influence of climate warming on 100-year precipitation events.

*Data availability.* The global estimates of 100-year return values and their confidence intervals on a 1°x 1° lat/lon grid from the EPS data and the REGEN, CHIRPS and PERSIANN observations, presented in this study, can be downloaded from http://dx.doi.org/10.17169/ refubium-39650. The operational ensemble forecast data from the ECMWF can be downloaded from https://apps.ecmwf.int/archive-catalogue/ ?type=cf&class=od&stream=enfo&expver=1 (ECMWF, 2023b) for the control run and from https://apps.ecmwf.int/archive-catalogue/?type= pf&class=od&stream=enfo&expver=1 (ECMWF, 2023f) for the perturbed runs. The user's affiliation needs to belong to an ECMWF member
state. The observational data sets are freely accessible from https://doi.org/10.25914/5ca4c380b0d44 (Contractor et al., 2020b) for REGEN, https://data.chc.ucsb.edu/products/CHIRPS-2.0/global_daily/netcdf/p25/ (Funk et al., 2014b) for CHIRPS and https://www.ncei.noaa.gov/ data/precipitation-persiann/access/ (Ashouri et al., 2015b) for PERSIANN.



*Author contributions.* Florian Ruff performed the analysis, produced the figures and drafted the manuscript. Both authors designed the study, discussed results and edited the manuscript.

*Competing interests.* The authors have no competing interests to declare.

*Acknowledgements.* The German Meteorological Service DWD and ECMWF are acknowledged for providing the operational IFS model data. This work used resources of the Deutsches Klimarechenzentrum (DKRZ), which is granted by its Scientific Steering Committee (WLA) under project ID bb1152. We are greatful to Felix Fauer and Henning Rust (both Freie Universität Berlin) for their helpful comments on the statistical approaches, and to all colleagues of the ClimXtreme Module D for their technical assistance.

*Financial support.* This study has been funded by the German Ministry of Education and Research (Bundesministerium für Bildung und Forschung, BMBF) in its strategy Research for Sustainability (FONA) in the framework of the ClimXtreme programme, sub-project A2 (MExRain, grant number 01LP1901C).



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
