# Peer review of "Global estimates of 100-year return values of daily precipitation from ensemble weather prediction data"

_EGUsphere, 2023_

## Referee Comment (RC1)

**Review**
*Global estimates of 100-year return values of daily precipitation from ensemble weather prediction data*

This paper's goal is to determine 100 year return values of annual maximum daily precipitation from a quasi-observational dataset (ensemble forecast) and compare these to 100-year return values from actual (semi-)observational products – both ground-based and satellite-based. In order to do so, generalised extreme value distributions are fitted to the annual maximum daily precipitation data (both the ensemble forecast and the observational data, it seems) and the 99% percentile value (1/100 years) is extracted. By comparing the results based on the large ensemble forecast and the observational data, the authors conclude that the estimated 100-year return levels are higher almost everywhere in the ensemble forecast data compared to observations, and the uncertainty range of the estimates, based on non-parametric bootstrapping, is smaller for the ensemble forecast data than for observations.

I think this study asks a relevant and useful question, and is thorough in its use of several observational products for comparison. The general choice of methods (extreme value theory) is appropriate for the purpose of the study. The manuscript is easy to read, albeit sometimes a bit too heavy on the details. I think this could become a good paper, but I see major methodical flaws that would need to be addressed first. In addition, the reproducibility is not guaranteed with the current level of detail in the method descriptions.

Below I first expand on these two major issues, followed by a list of more specific comments in order of appearance.

**Major methodical flaws**
The main, general problem is the lack of caution and thoroughness in the use of observational data. This manifests, firstly, in the lack of contextualisation of observational uncertainties for the data products used, and in the way statistical quantities based on very short observational records are presented and compared to EPS-ensemble results, without enough attention for the instrumental differences and the bias and uncertainties in observational results. As a consequence, the results and conclusions of the authors are likely to be misunderstood; they seem to suggest that observations show lower and more uncertain extreme precipitation than the ensemble forecast. In fact, the observations themselves are not what causes the difference, but the data processing and the "unfair" comparisons made.

1. My main concern is the use of observational data. As far as I can tell from the method description, observed 100 year return level estimates are based on the three observational datasets with lengths of 38 up to 65 values. These datasets are so short that a lot of caution is warranted when return periods much longer than the sample length are assessed. It is, even with much data, notoriously difficult to get the tail of extreme value distributions right. Furthermore, it is known that there is a systematic low bias when return levels are estimated from small samples (for shape parameters <0.5). This is all not taken into account enough in the generation and presentation of results.
   Furthermore, the authors compare results obtained from samples with N < 70 directly

to results obtained from samples with N > 1200, both for return levels as well as confidence intervals. The effects of sample size on this comparison are likely to be larger than any true systematic difference.

Lastly, as far as I can tell, confidence intervals for observational estimates are determined using non-parametric bootstrapping as for EPS data, i.e., re-sampling those small datasets. This produces a confidence interval, but if the original sample did not contain enough information to reflect the true underlying GEV tail accurately, any bootstrapped samples will not either. More importantly, comparing confidence intervals based on samples that differ in size by a factor of 15 (EPS:obs) is not very instructive, since the "real" differences are obscured by the differences caused by the different sample sizes.

Just for illustration, here is a tiny R-script and the resulting plot. The script computes the RL100s of two random GEV-distributed samples, one of size 50 and one of size 1200, using 1000 bootstrap resamples. The original 50 and 1200 samples come from the same prescribed GEV distribution as you can see. The plot, showing RL100 values and the bootstrapped 95% confidence intervals, reveals that the n = 50 sample shows a smaller RL100 with a larger bootstrapped confidence interval, despite coming from the same GEV distribution. The fact that the results in the paper also show this, can thus not be ascribed with certainty to anything else than the fact that the sample sizes are so different. (If the n=50 sample contains one or more very extreme values, the result is vastly different, hence the caution needed when dealing with such small samples.)

```
bigs = revd(1200, 0,1,0.1) #define random GEV-distr samples
smalls = revd(50, 0,1,0.1)

funz<-function(data){return(return.level(fevd(data), 100))} #
    function to determine RL100 from bootstrapped samples

dtp = data.frame("n" = c(rep(1200,1000), rep(50,1000)),
                      "RL100" = c(booter(bigs, funz,
    1000)$results,  booter(smalls,funz,1000)$results)))

**dataframe longformat with individual resampled RL100 values as a**
    function of sample size n
```

[Figure]

A possible way to address these issues, might be to generate independent subsamples of the EPS dataset of the length of the observational datasets, and use these subsamples to determine "quasi-observational" 100-year return levels (RL100), as well as a confidence range (the range spanned by all these subsamples). These can be directly compared to the RL100 values obtained from the observations. Subsequently, the EPS RL100 values based on these small subsamples, can be compared to other resampled EPS samples of progressively increasing sample size. In this way, the effects of the different data type/source can be separated from effects due to sample size.

In addition, a synthetic data study could be done to quantify the margin of error and potential bias in estimates based on small datasamples. E.g. one could generate 1000 independent samples of different lengths comparable to the observational datasets, based on random sampling from a known GEV distribution (or several, representative of several regions, for example). Then one could fit GEVs to these samples, and derive return levels from these GEVs. The "empirical" return levels as a function of sample size can be compared to the known true return level. See e.g. Zeder et al. (2023).

Naturally, there are other ways to make a justified comparison between the EPS data and the observational data, and to assess the sensitivities to the data properties. In

any case, the analysis should be much more careful around determining observed RL100 with the data at hand.

In the discussion it is indeed mentioned that the main cause for the differences in confidence interval is the sample size difference. This should get more attention earlier in the paper. If the main point of the paper is to show that the observational record length leads to biases, it should be restructured and backed up with more statistical analysis so that that point comes across more clearly.

2.  Also on the general use of the observational data I have some concerns. Firstly, REGEN comes with a mask reflecting quality/trust in the interpolated values (for many locations, there are hardly any measurements). It would be good to acknowledge this fact, perhaps simply use the quality masked dataset, or at least show where confidence in the REGEN data is generally low, regardless of sample size. Also, REGEN provides Rx1d values (annual precipitation max) (e.g. on https://climdex.org), computed in a way that tries to best reflect actual annual maximum precipitation. I'd recommend using this product directly, instead of computing it from daily values.

3.  Lastly, if CHIRPS and PERSIANN do not provide Rx1d products, they have to be calculated of course. In this procedure, the order of operations matters. As far as I can tell, the authors first regrid the data, after which they determine the annual maxima. This results in significantly lower values than the reverse order of operations due to spatial smoothing of extreme values. The preferred order of operations is this: first extraction of Rx1d, and then (conservative!) regridding. This order of operations conserves the intensity better. Given that the absolute magnitude of Rx1d return levels is central in the analysis, these details affect the results.

**Major content flaws**
4.  Essential information from the methods is missing, especially on the way observations are processed to obtain 100 year return levels and confidence intervals. I assumed the same way as the EPS data, but this is not clearly stated.

**Minor comments**
**General**
1.  Might it make sense to focus on precipitation over land? Two of the observational datasets have land coverage only anyway, and the ocean signal is so strong that it obscures the patterns over land due to the colourbar scaling being adjusted to the ocean mainly.

2.  Some references are not properly displayed, e.g. "Organization, 2009".

**Specific**
L6-10: In light of my general comments above, these conclusions might need revision.

L99-100: I would think that another reason to not use all 10 days of the forecast, even if the members were uncorrelated, is that you would have the same day in the ensemble multiple times, because there'd be overlap between forecasts made on day n and day n+1, which would introduce more dependence between individual values.

L127-128: "regridded"/"interpolation scheme MIR": more information is needed here. Is 1 by 1 degree the lowest resolution found in the original data? Is all data "upscaled" to lower resolution, or is some downscaled? What is MIR? What kind of interpolation scheme does it use (bilinear, conservative etc.)?

Sect. 2.2.1: as mentioned above, the data quality/confidence in REGEN is low for a large part of the global land, it would be good to mention and show this. I'd even recommend using the quality masked dataset.

L156: "interpolated": how, which scheme/method? Also, see major comment 3 on order of operations.

L167: "interpolated": see previous comment

L169: "set to 0": this is not necessary and does not introduce problems if the annual maximum is used only. See comment below for L211

Sections 2.2.1-.2.2.3: It would be useful if, for each of the 3 obs datasets, the details were summarised, such as the length and coverage (land only, <60N/S only etc) of the record.

Section 3: Major comment 4: the methods for return value and confidence interval determination from observational data is missing.

L191-194: suggest to remove: the results for the river catchments do not matter here.

L208-211: Also here it is not necessary to report on what Ruff & Pfahl found in their previous study.

L211: "50th and 90th percentiles". I wonder why this is done in this way, given that the analysis is about 100 year return levels of annual maximum precipitation, I do not think that the 50th and 90th percentile of daily precipitation really matter much. If there is good agreement at medium low percentiles, that does not guarantee that there is good agreement for the >99th percentile (where the annual max is location) as well. I would suggest assessing agreement in the distributions of Rx1d (the median and some measure of spread of the Rx1d distribution, for example).

L239: "for the observational datasets": it is not clear to me how the trend is computed - Supplementary Fig. 2 suggests it is also the 99.9th percentile trend. Does that mean the 99.9th percentile is determined for each year based on the 365 daily precipitation values? Might it make more sense to simply assess the presence of a trend in the timeseries of annual maxima for both EPS and observational data?

L256-257: "sufficient for the Fisher-Tippett theorem" I am not sure what is meant here. The number of blocks does not determine whether the data is GEV-distributed. It needs to be i.i.d. and max stable (and a large enough dataset to contain enough information).

We already know the data are more or less i.i.d. based on the correlation study, but testing max stability (e.g. qq plots) would be good.

L258: "estimating the location, scale and shape parameters": which method is used to estimate these?

L277-279: Perhaps L-moments (assuming MLE was used in the initial GEV fits) provides more robust results with less excessive parameters. For stationary GEVs, L-moments is generally the better way, see e.g. Hosking (1990).

Results section: using mm/day instead of mm would be more specific.

L285-292: This is a matter of taste perhaps, but I find the lengthy listing of absolute RL100 values not very useful. This also holds for lines L300-305, (partly) L326-355 (all the specific values can be seen in the figure, the text should contain interpretation rather than listing the values), L 370-381.

L293-298: In and of itself this section seems a bit lost. However, in L328, the authors mention "no clear pattern" in the relative CI-magnitudes, but there is: it is exactly the pattern in Fig. 5. This makes sense: where the spread is very large, and the tail long and thin, it is very difficult to estimate the tail percentiles. I would nonetheless suggest moving Fig. 5 to an appendix, and referring to that in L328.

L299-319: See major comment 1.

Fig. 4: The stippling is a bit too tight, so it becomes more like a gray haze.

L321-355: See major comment 1.

L337: What is p.p.?

Fig 7: I think the relative CI is the relevant quantity (Fig 6). Suggest to remove or move Fig 7 to appendix.

L364-365: It would be good to expand a bit on how the non-independence of the data in the tropical oceans and maritime continent might have affected the results.

L397: "substantially reduced in the EPS data": see major comment 1.

**References**
Zeder et al. (2023): https://agupubs.onlinelibrary.wiley.com/doi/pdf/10.1029/2023GL104090
Hosking (1990): https://www.jstor.org/stable/2345653

---

## Referee Comment (RC2)

**Review of "Global estimates of 100-year return values of daily precipitation from ensemble weather prediction data" by F. Ruff and S. Pfahl**

**General comments**

The authors develop a method to reduce the sampling error in estimates of very extreme daily precipitation, based on a new dataset consisting of a vast number of weather forecasts.

The manuscript is of a very high standard in terms of clarity, and contains a thorough description of data, methods and results. However, there is one science aspect that needs further consideration. While statistical uncertainty in estimates of extremes is reduced in this new dataset, the evidence on their bias is limited to precipitation amounts for normal weather types (1-in-2 day, and 1-in-10 day) rather than the extreme weather conditions producing the most intense rainfall. A fuller evaluation is required to prove ensemble forecast data from a relatively short period of real weather situations can provide more accurate estimates of very extreme precipitation.

If a revised version included more appropriate validation, and considered the extra points below, then publication as a NHESS Highlight article may be suitable since it has potential to make a substantial contribution to a subject of growing importance to society.

**Major comments**

**1.** The assessment of the new dataset versus observed values is described in lines 208-231, and Figures 2 and 3. It is limited to an evaluation of the 1-in-2 day, and 1-in-10 day precipitation, which concern the most common events producing precip at locations. However, the new dataset provides information on those very extreme events caused by rare atmosphere conditions, and no evidence is given on forecast bias for highly unusual weather events. As a consequence, little confidence can be attached to estimates of the 1-in-100 year return values from this new dataset.

The evaluation of the extreme right tail of forecast data is required. The observed data in this study are sufficiently long to contain relatively small statistical uncertainty for precip amounts exceeded every one year, up to perhaps once in 10 or 20 years. If the new dataset is consistent with such rare events (or can be calibrated to be more consistent), then more confidence could be placed in the new estimates of 1-in-100 year return values.

The last sentence in the Abstract may need revision after the model is validated.

**2.** There are many land areas in Figures 2 and 3 (b, c and d) which contain almost no information on bias due to the use of absolute values. Could the authors plot the relative differences (in %) between EPS and the three obs datasets in Figures 2 and 3 (or any new exhibits made as a result of comment 1 above)?

**3.** Could the authors include return value plots for a selection of locations representing their global results? For example, return period on the x-axis (a log scale, from 1 to 100 years, or more) and

return value on the y-axis, for EPS and the three observational datasets, and locations in interesting regions such as northern and southern Europe, west and east US, Arabian Peninsula, India, Brazil etc? This would be a very useful addition in Section 4 (after any bias correction is applied to EPS in Section 3, see major comment 1 above). Inclusion of Confidence Intervals for these representative locations, perhaps for EPS and REGEN, would be very useful for the reader too.

**4.** This comment is intended as a suggestion. Many of the maps are dominated by values over the ocean, which distracts the eye away from the changes over land. Given how this study is focused on 100 year return values, and these are most relevant to flooding on land, do the authors think it worthwhile to focus on land-only changes in all figures in the main manuscript? Maps with values over oceans could be included in a Supplementary section, if the authors wish to include them?

**5.** The added value from the eight maps in Figures 6 and 7 is quite limited, since EPS naturally has smaller sampling error than observed datasets due to its much longer record (1224 years versus a few decades). Perhaps 2 of these 8 maps would suffice, e.g. Figures 6a and b, leaving space for more locations to be examined in detail, discussed in point 3 above?

**Minor comments/corrections**

1. line 37: should the author be referred to as World Meteorological Organization, rather than 'Organization'? (With consistency in References section too.)

2. lines 115-116: could the text include the year of these ECMWF Cycles, to make them more meaningful for the reader?

3. line 144: change '135.000' to '135,000'

4. colours chosen in most figures: the authors choose different shades of red to represent wetter conditions, and blue to represent drier. In my experience, most researchers choose the reverse, I've become familiar with 'blue = wetter'. Would the authors consider reversing their colour scale, or are they more familiar with 'red = wetter'?

5. caption in Figure 1: "Mind the logarithmic colour scale" is acceptable spoken English, but not standard written English. This applies to captions for figures 5 and 6 too. A small tweak such as "Note the logarithmic colour scale" would be sufficient.

6. line 360: I recommend deleting 'hence, reduce flood risks' (not necessary and breaks the flow of sentence).

7. line 370 ff: it may be worth adding how these quoted precip values refer to averages over 1 deg grid cell areas? (For example, some northern Europe locations have recorded daily total far in excess of 100mm, due to events such as storm Bernd.)

---

## Author Response (AR1)

**Responses to the comments of Reviewer 1 and 2**

by Florian Ruff and Stephan Pfahl

We would like to thank both reviewers for their helpful comments and suggestions to improve the manuscript! This document addresses all comments of reviewer 1 and 2 in the sections 1 and 2, respectively. The reviewers' comments are repeated in black and italic with the corresponding abbreviation RC1 and RC2, respectively. Our responses are given in blue with the abbreviation AC (author comments).

**1 Comments of Reviewer 1**

*RC1: Review*

*Global estimates of 100-year return values of daily precipitation from ensemble weather prediction data*

*This paper's goal is to determine 100 year return values of annual maximum daily precipitation from a quasi-observational dataset (ensemble forecast) and compare these to 100-year return values from actual (semi-)observational products – both ground-based and satellite-based. In order to do so, generalised extreme value distributions are fitted to the annual maximum daily precipitation data (both the ensemble forecast and the observational data, it seems) and the 99% percentile value (1/100 years) is extracted. By comparing the results based on the large ensemble forecast and the observational data, the authors conclude that the estimated 100-year return levels are higher almost everywhere in the ensemble forecast data compared to observations, and the uncertainty range of the estimates, based on non-parametric bootstrapping, is smaller for the ensemble forecast data than for observations.*

*I think this study asks a relevant and useful question, and is thorough in its use of several observational products for comparison. The general choice of methods (extreme value theory) is appropriate for the purpose of the study. The manuscript is easy to read, albeit sometimes a bit too heavy on the details. I think this could become a good paper, but I see major methodical flaws that would need to be addressed first. In addition, the reproducibility is not guaranteed with the current level of detail in the method descriptions.*

*Below I first expand on these two major issues, followed by a list of more specific comments in order of appearance.*

AC:  Thank you very much for your helpful comments! We addressed the methodical comments in a revised version of the manuscript and described our methodical approaches in more detail.

**1.1  General comments of Reviewer 1**

*RC1: Major methodical flaws*

*The main, general problem is the lack of caution and thoroughness in the use of observational data. This manifests, firstly, in the lack of contextualisation of observational uncertainties for the data products used, and in the way statistical quantities based on very short observational records are presented and compared to EPS-ensemble results, without*

*enough attention for the instrumental differences and the bias and uncertainties in observational results. As a consequence, the results and conclusions of the authors are likely to be misunderstood; they seem to suggest that observations show lower and more uncertain extreme precipitation than the ensemble forecast. In fact, the observations themselves are not what causes the difference, but the data processing and the "unfair" comparisons made.*

AC: We agree with the reviewer that the differences between uncertainties of the observational and model-based estimates are due to the different lengths of the time series and thus can be explained by the statistical sampling. We did not want to claim that the model-based data set is superior to the observations in any other aspect. This is made clearer in the revised manuscript. However, we do not think that the comparison of return values from time series of different lengths in this context is an unfair comparison. The reduction in uncertainty due to sampling may appear a bit trivial, but it has practical consequences: Suppose that, for a practical problem (such as the construction of a dike), 100-year return values of daily precipitation are required (which is not very hypothetical). One then has the options to either estimate these return values based on observations, which are typically available from a relatively short period only, or from our model-based data set that provides longer time series. Our manuscript helps in judging the pros and cons of these approaches and provides information on the reduction of the statistical uncertainty, but also on the fact that the model-based estimates may be biased high compared to observations (see also below for the latter point). In this sense, the comparison is not unfair, but based on the actual data availability.

RC1: *1. My main concern is the use of observational data. As far as I can tell from the method description, observed 100 year return level estimates are based on the three observational datasets with lengths of 38 up to 65 values. These datasets are so short that a lot of caution is warranted when return periods much longer than the sample length are assessed. It is, even with much data, notoriously difficult to get the tail of extreme value distributions right. Furthermore, it is known that there is a systematic low bias when return levels are estimated from small samples (for shape parameters <0.5). This is all not taken into account enough in the generation and presentation of results.*

*Furthermore, the authors compare results obtained from samples with N < 70 directly to results obtained from samples with N > 1200, both for return levels as well as confidence intervals. The effects of sample size on this comparison are likely to be larger than any true systematic difference.*

*Lastly, as far as I can tell, confidence intervals for observational estimates are determined using non-parametric bootstrapping as for EPS data, i.e., re-sampling those small datasets. This produces a confidence interval, but if the original sample did not contain enough information to reflect the true underlying GEV tail accurately, any bootstrapped samples will not either. More importantly, comparing confidence intervals based on samples that differ in size by a factor of 15 (EPS:obs) is not very instructive, since the "real" differences are obscured by the differences caused by the different sample sizes.*

*Just for illustration, here is a tiny R-script and the resulting plot. The script computes the RL100s of two random GEV-distributed samples, one of size 50 and one of size 1200, using 1000 bootstrap resamples. The original 50 and 1200 samples come from the same prescribed GEV distribution as you can see. The plot, showing RL100 values and the bootstrapped 95% confidence intervals, reveals that the n = 50 sample shows a smaller RL100 with a larger bootstrapped confidence interval, despite coming from the same GEV distribution. The fact that the results in the paper also show this, can thus not be ascribed with certainty to anything else than the fact that the sample sizes are so different. (If the n=50 sample contains one or more very extreme values, the result is vastly different, hence the caution needed when dealing with such small samples.)*

*bigs = revd(1200, 0,1,0.1) #define random GEV-distr samples*

*smalls = revd(50, 0,1,0.1)*

*funz<-function(data){return(return.level(fevd(data), 100))} # function to determine RL100 from bootstrapped samples*

*dtp = data.frame("n" = c(rep(1200,1000), rep(50,1000)), "RL100" = c(booter(bigs, funz, 1000)$results, booter(smalls,funz,1000)$results)))*

*#dataframe longformat with individual resampled RL100 values as a function of sample size n*

*A possible way to address these issues, might be to generate independent subsamples of the EPS dataset of the length of the observational datasets, and use these subsamples to determine "quasi-observational" 100-year return levels (RL100), as well as a confidence range (the range spanned by all these subsamples). These can be directly compared to the RL100 values obtained from the observations. Subsequently, the EPS RL100 values based on these small subsamples, can be compared to other resampled EPS samples of progressively increasing sample size. In this way, the effects of the different data type/source can be separated from effects due to sample size.*

*In addition, a synthetic data study could be done to quantify the margin of error and potential bias in estimates based on small datasamples. E.g. one could generate 1000 independent samples of different lengths comparable to the observational datasets, based on random sampling from a known GEV distribution (or several, representative of several regions, for example). Then one could fit GEVs to these samples, and derive return levels from these GEVs. The "empirical" return levels as a function of sample size can be compared to the known true return level. See e.g. Zeder et al. (2023).*

*Naturally, there are other ways to make a justified comparison between the EPS data and the observational data, and to assess the sensitivities to the data properties. In any case, the analysis should be much more careful around determining observed RL100 with the data at hand.*

*In the discussion it is indeed mentioned that the main cause for the differences in confidence interval is the sample size difference. This should get more attention earlier in the paper. If the main point of the paper is to show that the observational record length leads to biases, it should be restructured and backed up with more statistical analysis so that that point comes across more clearly.*

AC: See our response above for the aspect of comparison of return value estimate from records with different length. As outlined there, we agree with the reviewer that most of the differences can be explained by the different sampling, but, from a practical standpoint, it cannot be avoided that observational time series are short and observational estimates of 100-year return values are thus highly uncertain. The main point of our analysis is that this sampling uncertainty can be reduced with model-based data, albeit they may introduce additional issues (such as a model bias). In this sense, it is impossible to evaluate „true systematic differences" between the data set, as much longer observational time series would be required. In such an ideal world with 1000 years of observational data, our model-based approach would most likely not be required, but since we don't live in such a world, we think it is worth to compare the two approaches including their different sampling properties (lengths of records).

Nevertheless, we agree that it is a very good suggestion to probe the effect of sample size by subsampling the EPS data. We included such an analysis in the revised manuscript. Our results (see Figure below) show that the sampling effect can indeed explain the reduced uncertainty in the EPS estimates (as expected), but not (or only to a very small degree) the

[Figure]

Fig. 1: Boxplot distribution of 1000 return value estimates of a 100-year event based on different sample sizes of block maxima for a single grid point over Germany (52°N, 13°E). The red bars indicate the 95% confidence interval. The blue line indicates the 100-year estimate from the largest sample size of 1200.

systematic overestimation, which is thus more likely related to a model (or observational) bias. We adapted the discussion in the manuscript accordingly and also emphasised the importance of the observational record length much earlier and more prominently in the paper.

RC1: *2. Also on the general use of the observational data I have some concerns. Firstly, REGEN comes with a mask reflecting quality/trust in the interpolated values (for many locations, there are hardly any measurements). It would be good to acknowledge this fact, perhaps simply use the quality masked dataset, or at least show where confidence in the REGEN data is generally low, regardless of sample size. Also, REGEN provides Rx1d values (annual precipitation max) (e.g. on https://climdex.org), computed in a way that tries to best reflect actual annual maximum precipitation. I'd recommend using this product directly, instead of computing it from daily values.*

AC: Thanks for these suggestions. We used the quality mask in REGEN observation related figures but decided to skip the Rx1d product from Climdex. On the one hand, we were not able to find more detailed information about the determination process of Rx1d precipitation data from Climdex and, thus, are not able to explain the difference to our data set. On the other hand, we want to avoid to use interpolated annual maxima but instead calculate the annual maxima of precipitation averaged over grid boxes as explained in our response to the following comment.

RC1: *3. Lastly, if CHIRPS and PERSIANN do not provide Rx1d products, they have to be calculated of course. In this procedure, the order of operations matters. As far as I can tell, the authors first regrid the data, after which they determine the annual maxima. This results in significantly lower values than the reverse order of operations due to spatial smoothing of extreme values. The preferred order of operations is this: first extraction of Rx1d, and then (conservative!) regridding. This order of operations conserves the intensity better. Given that the absolute magnitude of Rx1d return levels is central in the analysis, these details affect the results.*

AC: We do not agree that there is a preferred order of these operations. Rather, the two different approaches refer to different physical quantities. On the one hand, determining the return values on a high-resolution grid (or directly on the station level) and then interpolating to other locations provides estimates of local return values at locations where no direct observations are available. This is relevant, e.g., for local flash floods. On the other hand, first regridding the data and then determining the return values provides estimates of extremes of area-averaged precipitation. This is relevant, e.g., for flooding in larger rivers that integrate the precipitation over a large catchment (as we also argue in Ruff and Pfahl, 2023). Here, we decided to focus on the latter, since this is also what models provide (they

simulate the area-average precipitation over a model grid box and not a point estimate). Remapping both precipitation data sets to the same grid thus allows for a fair comparison. We explained this more explicitly in the revised manuscript.

*RC1: Major content flaws*

*4. Essential information from the methods is missing, especially on the way observations are processed to obtain 100 year return levels and confidence intervals. I assumed the same way as the EPS data, but this is not clearly stated.*

AC: The determination of 100-year return values and confidence intervals from the observations are the same as for the EPS data. This is stated in the revised manuscript.

*RC1: Minor comments*

*General*

*1. Might it make sense to focus on precipitation over land? Two of the observational datasets have land coverage only anyway, and the ocean signal is so strong that it obscures the patterns over land due to the colourbar scaling being adjusted to the ocean mainly.*

*2. Some references are not properly displayed, e.g. "Organization, 2009".*

AC: Thank you for this suggestion. Such a focus on only land surfaces is also more relevant for future applications of the results. We adapted the figures in the revised manuscript to only show values over land.

The wrongly displayed references are corrected.

**1.2 Specific comments of Reviewer 1**

*RC1: L6-10: In light of my general comments above, these conclusions might need revision.*

AC: We adapted the abstract accordingly and emphasised the sampling effect in explaining the reduced uncertainty. In addition, we also commented more critically on the potential model bias.

*RC1: L99-100: I would think that another reason to not use all 10 days of the forecast, even if the members were uncorrelated, is that you would have the same day in the ensemble multiple times, because there'd be overlap between forecasts made on day n and day n+1, which would introduce more dependence between individual values.*

AC: Thank you for that comment. Yes, the forecast for day n and day n+1 would be quite similar, which would decrease the effective sample size, when there is an extreme event during both forecast days. We added a sentence mentioning this in the revised manuscript.

*RC1: L127-128: "regridded"/"interpolation scheme MIR": more information is needed here. Is 1 by 1 degree the lowest resolution found in the original data? Is all data "upscaled" to lower resolution, or is some downscaled? What is MIR? What kind of interpolation scheme does it use (bilinear, conservative etc.)?*

AC: The original EPS precipitation data have a higher resolution for the entire period. By downloading the data on a coarser grid (here 1°x1°), the Meteorological Interpolation and Regridding (MIR) scheme by the ECMWF "upscale" the data as needed. This is done by linear interpolation based on a triangular mesh, no conservative remapping, which is a drawback for the current study. The MIR scheme may not reproduce the most accurate

results of an upscaling from a higher resolution grid to a lower resolution grid. More details can be found in ECMWF (2023). We included these information in the revised manuscript.

*RC1: Sect. 2.2.1: as mentioned above, the data quality/confidence in REGEN is low for a large part of the global land, it would be good to mention and show this. I'd even recommend using the quality masked dataset.*

AC: Thank you, we added the information of the quality mask as stippling in several figures, that contain REGEN observations, and included information in the description of REGEN observations.

*RC1: L156: "interpolated": how, which scheme/method? Also, see major comment 3 on order of operations.*

*RC1: L167: "interpolated": see previous comment*

AC: The data are available on a 0.25x0.25 degree resolution and are averaged over a 1x1 degree box. We changed this phrase, as the word "interpolated" might be misleading.

*RC1: L169: "set to 0": this is not necessary and does not introduce problems if the annual maximum is used only. See comment below for L211*

AC: The reviewer is right that this does not lead to problems for the estimation of 100-year return values. However, it does play a role for validating the bias of the EPS data, as it changes the values of the percentiles. As this is the intention behind setting all missing values to 0, we still keep this operation for the PERSIANN data.

*RC1: Sections 2.2.1-.2.2.3: It would be useful if, for each of the 3 obs datasets, the details were summarised, such as the length and coverage (land only, <60N/S only etc) of the record.*

AC: We included a small table that summarises the most important details of the observational data sets in the revised manuscript.

*RC1: Section 3: Major comment 4: the methods for return value and confidence interval determination from observational data is missing.*

AC: The method is identical for the EPS and observational data. We added a sentence regarding this in Section 3.2.

*RC1: L191-194: suggest to remove: the results for the river catchments do not matter here.*

*RC1: L208-211: Also here it is not necessary to report on what Ruff & Pfahl found in their previous study.*

AC: We think that a short summary of the statistical evaluation of the data set performed by Ruff and Pfahl (2023) is helpful as it provides some context for the present statistical evaluation. We thus kept these sentences, but reduced their length.

*RC1: L211: "50th and 90th percentiles". I wonder why this is done in this way, given that the analysis is about 100 year return levels of annual maximum precipitation, I do not think that the 50th and 90th percentile of daily precipitation really matter much. If there is good agreement at medium low percentiles, that does not guarantee that there is good agreement for the >99th percentile (where the annual max is location) as well. I would suggest assessing agreement in the distributions of Rx1d (the median and some measure of spread of the Rx1d distribution, for example).*

AC: Thank you, we switched to higher percentiles (99th and 99.99th) in order to evaluate the agreement for (more or less) yearly maxima and even more intense events.

RC1: L239: *"for the observational datasets": it is not clear to me how the trend is computed - Supplementary Fig. 2 suggests it is also the 99.9th percentile trend. Does that mean the 99.9th percentile is determined for each year based on the 365 daily precipitation values? Might it make more sense to simply assess the presence of a trend in the timeseries of annual maxima for both EPS and observational data?*

AC: Thank you. Yes, the 99.9th percentile was based on the 365 daily precipitation values and the use of yearly maxima is more reasonable to assess the presence of trends. We changed this in the revised manuscript.

RC1: L256-257: *"sufficient for the Fisher-Tippett theorem" I am not sure what is meant here. The number of blocks does not determine whether the data is GEV-distributed. It needs to be i.i.d. and max stable (and a large enough dataset to contain enough information).*

*We already know the data are more or less i.i.d. based on the correlation study, but testing max stability (e.g. qq plots) would be good.*

AC: We rewrote this sentence but did not test the precipitation data for max stability as we are not entirely certain what is exactly meant. We did not show that each ensemble member has a similar daily precipitation distribution a each grid point because this similarity is shown in Ruff & Pfahl (2023), at least for central Europe. However, we do not expect a different behaviour for other regions as mainly parameterisations and initial conditions are changed throughout the members. If „max-stable" means that the distribution of the daily precipitation keeps similar for the selected set of block maxima, we do not see this as a problem but it is also not likely because the block maxima are GEV distributed and the precipitation data are (if anything) gamma distributed.

RC1: L258: *"estimating the location, scale and shape parameters": which method is used to estimate these?*

AC: The estimation is performed by the maximum likelihood approach. We included this in the revised manuscript.

RC1: L277-279: *Perhaps L-moments (assuming MLE was used in the initial GEV fits) provides more robust results with less excessive parameters. For stationary GEVs, L-moments is generally the better way, see e.g. Hosking (1990).*

AC: Thank you for this suggestion. Yes, MLE is used in the GEV fits as it is asymptotically unbiased and the variance of the estimated parameters is also smaller than for L-moments, especially for large samples (see Davison and Huser, 2015, and Coles et al., 2001). L-moments might have an advantage when the shape parameter is smaller 0 (Martins & Stedinger, 2000), which is the case for 1% of grid points in our study. Therefore, we stick to this MLE method.

RC1: *Results section: using mm/day instead of mm would be more specific.*

AC: Changed.

RC1: L285-292: *This is a matter of taste perhaps, but I find the lengthy listing of absolute RL100 values not very useful. This also holds for lines L300-305, (partly) L326-355 (all the specific values can be seen in the figure, the text should contain interpretation rather than listing the values), L 370-381.*

AC: We reduced the information to the most important values.

RC1: L293-298: *In and of itself this section seems a bit lost. However, in L328, the authors mention "no clear pattern" in the relative CI-magnitudes, but there is: it is exactly the pattern in Fig. 5. This makes sense: where the spread is very large, and the tail long and thin, it is*

*very difficult to estimate the tail percentiles. I would nonetheless suggest moving Fig. 5 to an appendix, and referring to that in L328.*

AC: We added a few sentences on this connection to Fig. 5 and moved Fig. 5 to the supplement.

*RC1: L299-319: See major comment 1.*

AC: Adapted based on the results of the further analyses.

*RC1: Fig. 4: The stippling is a bit too tight, so it becomes more like a gray haze.*

AC: We changed it to a wider stippling.

*RC1: L321-355: See major comment 1.*

AC: We rewrote this paragraph based on the further analyses.

*RC1: L337: What is p.p.?*

AC: The difference of two percentages is here presented in percentage point (p.p.). We included an explanation of this unit in the revised manuscript.

*RC1: Fig 7: I think the relative CI is the relevant quantity (Fig 6). Suggest to remove or move Fig 7 to appendix.*

AC: We moved the difference figures between EPS and CHIRPS as well as between EPS and PERSIANN to the supplement for both Fig. 6 and Fig. 7, as they show similar results to the differences between EPS and REGEN and we reduce the amount of figures. Still, we think the absolute uncertainty is interesting as it increases over some regions, which might not be obvious, so that we kept 2 panels of Fig. 7.

*RC1: L364-365: It would be good to expand a bit on how the non-independence of the data in the tropical oceans and maritime continent might have affected the results.*

AC: Thank you, we mentioned the limitation of non-independence between ensemble members in more detail in the revised manuscript.

*RC1: L397: "substantially reduced in the EPS data": see major comment 1.*

AC: See our response to major comment 1.

**2 Comments of Reviewer 2**

*RC2: The authors develop a method to reduce the sampling error in estimates of very extreme daily precipitation, based on a new dataset consisting of a vast number of weather forecasts.*

*The manuscript is of a very high standard in terms of clarity, and contains a thorough description of data, methods and results. However, there is one science aspect that needs further consideration. While statistical uncertainty in estimates of extremes is reduced in this new dataset, the evidence on their bias is limited to precipitation amounts for normal weather types (1-in-2 day, and 1-in-10 day) rather than the extreme weather conditions producing the most intense rainfall. A fuller evaluation is required to prove ensemble forecast*

*data from a relatively short period of real weather situations can provide more accurate estimates of very extreme precipitation.*

*If a revised version included more appropriate validation, and considered the extra points below, then publication as a NHESS Highlight article may be suitable since it has potential to make a substantial contribution to a subject of growing importance to society.*

AC:  Thank you very much for your helpful comments! We investigated the bias for extreme precipitation events and took your comments into account in a revised version of the manuscript.

**2.1   Major comments of Reviewer 2**

*RC2: 1. The assessment of the new dataset versus observed values is described in lines 208-231, and Figures 2 and 3. It is limited to an evaluation of the 1-in-2 day, and 1-in-10 day precipitation, which concern the most common events producing precip at locations. However, the new dataset provides information on those very extreme events caused by rare atmosphere conditions, and no evidence is given on forecast bias for highly unusual weather events. As a consequence, little confidence can be attached to estimates of the 1-in-100 year return values from this new dataset.*

*The evaluation of the extreme right tail of forecast data is required. The observed data in this study are sufficiently long to contain relatively small statistical uncertainty for precip amounts exceeded every one year, up to perhaps once in 10 or 20 years. If the new dataset is consistent with such rare events (or can be calibrated to be more consistent), then more confidence could be placed in the new estimates of 1-in-100 year return values.*

*The last sentence in the Abstract may need revision after the model is validated.*

AC:  Thank you, we evaluated the data set with regard to higher percentiles that are representative of (almost) yearly maxima (99th percentile) and (almost) 30-year precipitation events (99.99th percentile) in comparison to the observational data sets. Although there are larger differences, especially for higher percentiles, between the EPS and observational data sets, we are critical about a bias correction of the data set. The bias largely differs from region to region and observational data sets do not agree over tropical and subtropical areas. We added a short statement regarding this in the revised manuscript.

The abstract is adapted as well.

*RC2: 2. There are many land areas in Figures 2 and 3 (b, c and d) which contain almost no information on bias due to the use of absolute values. Could the authors plot the relative differences (in %) between EPS and the three obs datasets in Figures 2 and 3 (or any new exhibits made as a result of comment 1 above)?*

AC:  Thank you, we changed the colour scales of the figures in order to better evaluate the lower values and added an evaluation of the relative differences of the specific percentiles.

*RC2: 3. Could the authors include return value plots for a selection of locations representing their global results? For example, return period on the x-axis (a log scale, from 1 to 100 years, or more) and return value on the y-axis, for EPS and the three observational datasets, and locations in interesting regions such as northern and southern Europe, west and east US, Arabian Peninsula, India, Brazil etc? This would be a very useful addition in Section 4 (after any bias correction is applied to EPS in Section 3, see major comment 1 above). Inclusion of Confidence Intervals for these representative locations, perhaps for EPS and REGEN, would be very useful for the reader too.*

AC: We added return value plots for a few locations with different behaviour between the EPS estimates and the observational estimates in the supplement and commented on that in the revised manuscript.

RC2: 4. This comment is intended as a suggestion. Many of the maps are dominated by values over the ocean, which distracts the eye away from the changes over land. Given how this study is focused on 100 year return values, and these are most relevant to flooding on land, do the authors think it worthwhile to focus on land-only changes in all figures in the main manuscript? Maps with values over oceans could be included in a Supplementary section, if the authors wish to include them?

AC: Thank you for this suggestion. Indeed, the focus of the results can be on the land areas as they are also more important for public flood protection systems. We changed the figures in the revised manuscript to land-only.

RC2: 5. The added value from the eight maps in Figures 6 and 7 is quite limited, since EPS naturally has smaller sampling error than observed datasets due to its much longer record (1224 years versus a few decades). Perhaps 2 of these 8 maps would suffice, e.g. Figures 6a and b, leaving space for more locations to be examined in detail, discussed in point 3 above?

AC: We agree that the added value might also be partly obvious. However, it is interesting to see that the assumption of a reduced error due to the smaller sample size in the EPS is really shown in most of the areas. Additionally, there are also at least some smaller regions where the uncertainty is larger in the EPS estimates, especially when considering absolute values, which might not be obvious. We reduced the number of panels and just show the differences to the REGEN observations, as the differences to the other two observational data sets (shifted to the supplement) are mostly similar.

**2.2 Minor comments of Reviewer 2**

RC2: 1. line 37: should the author be referred to as World Meteorological Organization, rather than 'Organization'? (With consistency in References section too.)

AC: This reference is corrected.

RC2: 2. lines 115-116: could the text include the year of these ECMWF Cycles, to make them more meaningful for the reader?

AC: We included the year of the changes.

RC2: 3. line 144: change '135.000' to '135,000'

AC: Is changed.

RC2: 4. colours chosen in most figures: the authors choose different shades of red to represent wetter conditions, and blue to represent drier. In my experience, most researchers choose the reverse, I've become familiar with 'blue = wetter'. Would the authors consider reversing their colour scale, or are they more familiar with 'red = wetter'?

AC: Thank you, we changed the colours that refer to wetter conditions into turquoise and that refer to dryer conditions into brown in order to reduce confusion with other red/blue colour scales of the manuscript figures.

*RC2: 5. caption in Figure 1: "Mind the logarithmic colour scale" is acceptable spoken English, but not standard written English. This applies to captions for figures 5 and 6 too. A small tweak such as "Note the logarithmic colour scale" would be sufficient.*

AC: Thank you, the sentences in the figure captions are changed.

*RC2: 6. line 360: I recommend deleting 'hence, reduce flood risks' (not necessary and breaks the flow of sentence).*

AC: Has been removed.

*RC2: 7. line 370 ff: it may be worth adding how these quoted precip values refer to averages over 1 deg grid cell areas? (For example, some northern Europe locations have recorded daily total far in excess of 100mm, due to events such as storm Bernd.)*

AC: We added a note in the sentence.

**References**

Coles, S., Bawa, J., Trenner, L., and Dorazio, P.: An Introduction to Statistical Modeling of Extreme Values. Vol. 208. London: Springer, 2001.

Davison, A. C. and Huser, R.: Statistics of Extremes, Annual Reviews of Statistics and its Application, 2(1), 203–235, https://doi.org/10.1146/annurev-statistics-010814-020133, 2015.

ECMWF: MARS interpolation with MIR, last access: 2024/03/12, https://confluence.ecmwf.int/display/UDOC/MARS+interpolation+with+MIR, 2023.

Martins, E. S. and Stedinger, J. R.: Generalized maximum-likelihood generalized extreme-value quantile estimators for hydrologic data, Water Resources Research, 36(3), 737-744, https://doi.org/10.1029/1999WR900330, 2000.

Ruff, F. and Pfahl, S.: What distinguishes 100-year precipitation extremes over Central European river catchments from more moderate extreme events?, Weather and Climate Dynamics, 4, 427–447, https://doi.org/10.5194/wcd-4-427-2023, 2023.

---

## Referee Report (RR1)

**Review of "Global estimates of 100-year return values of daily precipitation from ensemble weather prediction data" by F. Ruff and S. Pfahl**

**General**

**Overview of manuscript**: estimates of extreme values based on observations contain large uncertainties due to (i) errors due to finite length of record, (ii) spatial inhomogeneities in data, and (iii) trends due to climate change. I would add a fourth source of uncertainty, namely a user's subjective choices for EV modelling (type of extreme value distribution, and parameter fitting methods). This article shows how ensemble forecasts can avert the set of problems affecting observation-based estimates, however, the forecasted values may contain a new source of uncertainty due to model errors in representing precipitation processes.

I would like to thank the authors for their responses to reviewer #2 of the original manuscript. The revised manuscript is of a high standard in terms of science and clarity and contains some very interesting new information. It is suitable for publication, though might be improved by considering the comments below, before finalising the manuscript.

**Main comment**

Both the Abstract and the final section could be improved by a full, clear statement on all the problems with observational-based estimates of extreme quantiles, and on the other hand, a new potential problem concerning ensemble forecast bias, as described in "Overview of manuscript" above (which is largely distilled from the Introduction of manuscript).

These two sections might be improved by emphasising how future work on understanding the cause of large-scale bias between EPS and observations at extreme quantiles, especially in the tropics, is required.

**Minor comments/corrections**

Lines 44-50: the authors may wish to include a fourth limitation: users make subjective choices for EV modelling to extrapolate records to longer return periods.

Line 50: add a reference after the text "climate change", e.g. Fischer et al. (2014), or IPCC sixth assessment report?

Lines 156, 167 and 177: change "31th" to "31st"

Line 264: replace "Figs 2 and 2" with "Figs 2 and S2"

Line 274: most statistical software packages include *bias corrected* MLE methods. For example, the "mle.tools" package in R provides users with much better parameter fitting than basic MLE. Statisticians have worked on the problem of MLE bias for decades (mle.tools is based on a paper by Cox and Snell from 1968) yet their solutions are rarely used in meteorology. The bias corrected version of MLE will produce similar results as the basic MLE for 66-year REGEN records extrapolated

to 100 yr return levels, so this comment is not material for this article, but perhaps the authors might get benefits from using bias corrected MLE in future work?

Line 337: this is usually described as a thick tail, rather than a "long and *thin* tail".

**Reference**

Fischer, E. M., J. Sedláček, E. Hawkins, and R. Knutti (2014), Models agree on forced response pattern of precipitation and temperature extremes, Geophys. Res. Lett., 41, 8554–8562, doi:10.1002/2014GL062018.

---

## Author Response (AR2)

**Responses to the comments of Reviewer 2**

by Florian Ruff and Stephan Pfahl

We would like to thank the reviewer for the additional helpful comments and suggestions to improve the manuscript! This document addresses all comments of the reviewer. The reviewers' comments are repeated in black and italic with the abbreviation RC. Our responses are given in blue with the abbreviation AC (author comments).

**1   Comments of the Reviewer**

RC: ***Overview of manuscript****: estimates of extreme values based on observations contain large uncertainties due to (i) errors due to finite length of record, (ii) spatial inhomogeneities in data, and (iii) trends due to climate change. I would add a fourth source of uncertainty, namely a user's subjective choices for EV modelling (type of extreme value distribution, and parameter fitting methods). This article shows how ensemble forecasts can avert the set of problems affecting observation-based estimates, however, the forecasted values may contain a new source of uncertainty due to model errors in representing precipitation processes.*

*I would like to thank the authors for their responses to reviewer #2 of the original manuscript. The revised manuscript is of a high standard in terms of science and clarity and contains some very interesting new information. It is suitable for publication, though might be improved by considering the comments below, before finalising the manuscript.*

AC: Thank you very much for your additional helpful comments! We addressed all comments in a revised version of the manuscript.

**1.1   Major comments**

RC: *Both the Abstract and the final section could be improved by a full, clear statement on all the problems with observational-based estimates of extreme quantiles, and on the other hand, a new potential problem concerning ensemble forecast bias, as described in "Overview of manuscript" above (which is largely distilled from the Introduction of manuscript).*

*These two sections might be improved by emphasising how future work on understanding the cause of large-scale bias between EPS and observations at extreme quantiles, especially in the tropics, is required.*

AC: Thank you very much for this comment. We emphasised the problems of observation-based estimates and the model bias in both the abstract and the discussion and added a sentence on the importance of future work on understanding the model bias.

**1.2  Minor comments**

*RC: Lines 44-50: the authors may wish to include a fourth limitation: users make subjective choices for EV modelling to extrapolate records to longer return periods.*

AC:  Thank you, we added a sentence regarding this additional limitation.

*RC: Line 50: add a reference after the text "climate change", e.g. Fischer et al. (2014), or IPCC sixth assessment report?*

AC:  Thank you, we added the reference.

*RC: Lines 156, 167 and 177: change "31th" to "31st"*

AC:  Is changed.

*RC: Line 264: replace "Figs 2 and 2" with "Figs 2 and S2"*

AC:  Thank you, has been changed.

*RC: Line 274: most statistical software packages include bias corrected MLE methods. For example, the "mle.tools" package in R provides users with much better parameter fitting than basic MLE. Statisticians have worked on the problem of MLE bias for decades (mle.tools is based on a paper by Cox and Snell from 1968) yet their solutions are rarely used in meteorology. The bias corrected version of MLE will produce similar results as the basic MLE for 66-year REGEN records extrapolated to 100 yr return levels, so this comment is not material for this article, but perhaps the authors might get benefits from using bias corrected MLE in future work?*

AC:  Thank you very much for the additional tool box and the information about the bias corrected MLE methods. We will definitely keep this in mind and consider a possible use in future work.

*RC: Line 337: this is usually described as a thick tail, rather than a "long and thin tail".*

AC:  Thank you, we changed the description.

**References**

Fischer, E. M., Sedláček, J., Hawkins, E., & Knutti, R. (2014). Models agree on forced response pattern of precipitation and temperature extremes. *Geophysical Research Letters*, *41*(23), 8554-8562.